# U-MATH: A University-Level Benchmark for Evaluating Mathematical Skills in LLMs

## Abstract

The current evaluation of mathematical skills in LLMs is limited, as existing benchmarks are relatively small, primarily focus on elementary and high-school problems, or lack diversity in topics. Additionally, the inclusion of visual elements in tasks remains largely under-explored.

To address these gaps, we introduce **U-MATH**, a novel benchmark of **1,100** unpublished open-ended university-level problems sourced from teaching materials. It is balanced across six core subjects, with **20% of multimodal problems**. Given the open-ended nature of U-MATH problems, we employ an LLM to judge the correctness of generated solutions. To this end, we release $\mu$-**MATH**, an dataset to evaluate the LLMs' capabilities in judging solutions.

The evaluation of general domain, math-specific, and multimodal LLMs highlights the challenges presented by U-MATH. Our findings reveal that LLMs achieve a maximum accuracy of only 63% on text-based tasks, with even lower 45% on visual problems. The solution assessment proves challenging for LLMs, with the best LLM judge having an F1-score of 80% on $\mu$-MATH

During review, we publish the U-MATH and $\mu$-MATH datasets on OSF.[1].

---

**Example: Differential Calculus.**

**U-MATH Problem:**
The function $s(t) = 2 \cdot t^3 - 3 \cdot t^2 - 12 \cdot t + 8$ represents the position of a particle traveling along a horizontal line.
1. Find the velocity and acceleration functions.
2. Determine the time intervals when the object is slowing down or speeding up.

- - - - - - - - - - - - - - - - - - - - - - - - - - - - - - - - - - - - - - - - - - - - - - - - - -

**Reference Solution (shortened):**

The velocity is $v(t) = s'(t) = \boxed{6 \cdot t^2 - 6 \cdot t - 12}$, zeros of the $v(t)$ are $t = -1, 2$.

The acceleration is $a(t) = v'(t) = \boxed{12 \cdot t - 6}$, zero of the $a(t)$ is $t = \frac{1}{2}$.

It speeds up when $v(t)$ and $a(t)$ have the same sign, and slows down when opposite.

| Interval | $v(t)$ | $a(t)$ | Behavior |
|---|---|---|---|
| $(-\infty, -1)$ | $> 0$ | $< 0$ | Slowing down |
| $(-1, \frac{1}{2})$ | $< 0$ | $< 0$ | Speeding up |
| $(\frac{1}{2}, 2)$ | $< 0$ | $> 0$ | Slowing down |
| $(2, \infty)$ | $> 0$ | $> 0$ | Speeding up |

Accounting for non-negative time, speed up on $\boxed{(0, {}^1/_2) \text{ and } (2, \infty)}$, slow down on $\boxed{({}^1/_2, 2)}$.

Figure 1: U-MATH covers university-level topics and require multiple steps to solve. A random sample is provided; reference solution is shortened. In this example, common error is overlooking time non-negativity.

## 1 Introduction

Mathematical reasoning is a fundamental domain for assessing the true capabilities of Large Language Models (LLMs) to reason (Ahn et al., 2024). While existing benchmarks like GSM8K (Cobbe et al.,

---

[1] https://osf.io/jpsa4/?view_only=d588b9fa862345cb98ccf7238a157cea

2021) and MATH (Hendrycks et al., 2021) provide valuable insights, they primarily focus on school-level mathematics. This leaves a significant gap in understanding how LLMs perform on more advanced, university-level problems. Moreover, these benchmarks are becoming saturated, as GPT-4, using advanced prompting techniques, has achieved over 92% success rate on GSM8K and 80% on MATH (Achiam et al., 2023).

Recent works, such as CHAMP (Mao et al., 2024) and MathOdyssey (Fang et al., 2024), aim to introduce more challenging problems but are limited in size (<400 samples) and lack comprehensive topic coverage. The most challenging problems stem from school-level competitions or olympiads, missing the crucial middle ground of university-level coursework that reflects academic demands.

Furthermore, there is a growing interest in assessing multi-modal LLMs' abilities to perform mathematical reasoning involving visual elements (Ahn et al., 2024). Large datasets like MathVista (Lu et al., 2023), We-Math (Qiao et al., 2024), or MathVerse (Zhang et al., 2024) provide an extensive set of (mostly) visual tasks but may lack university-level problems and often rely on multiple-choice validation, leading to easier problems and faster saturation of benchmarks.

In turn, evaluating complex free-form answers remains a significant challenge for the field (Hendrycks et al., 2021). Current methods often rely on LLM judges to assess problems, which introduces potential biases and inconsistencies (Zheng et al., 2023). Errors introduced by automatic evaluators are often overlooked in popular benchmarks. This oversight makes it impossible to account for judge biases, which detracts from the reliability of the evaluation results.

Recent studies also indicate that evaluation of mathematical solutions is a demanding task (Zeng et al., 2023; Xia et al., 2024) and that an LLM's ability to judge mathematical solutions is correlated with its problem-solving performance (Stephan et al., 2024), further signifying the importance of evaluations designed to asses the evaluators themselves — also called meta-evaluations.

Popular datasets for the task of mathematical meta-evaluation are PRM800K (Lightman et al., 2023), MR-GSM8K (Zeng et al., 2023) and MR-MATH (Xia et al., 2024). However, these are all based on the GSM8K and MATH datasets, still leaving a gap in meta-evaluations for university-level problems.

Aiming to bridge these gaps and provide a comprehensive evaluation of LLMs' mathematical capabilities, we introduce **U-MATH** (*U*niversity *Math*) and a supplementary meta-evaluation dataset, which we refer to as **$\mu$-MATH** (*M*eta *U-MATH*). Our main contributions are:

1. **U-MATH Benchmark** (Section 3): We open-source a set of 1,100 of university-level problems collected from actual coursework with final answers and solutions. About 20% of problems require image *understanding* to be solved. The text-only part of the benchmark is balanced across 6 key subjects: Precalculus, Algebra, Differential Calculus, Integral Calculus, Multivariable Calculus, and Sequences&Series.

2. **$\mu$-MATH Meta-Evaluation Benchmark** (Section 3.3): Additionally, we introduce a set of 1084 meta-evaluation tasks sourced from U-MATH problems and designed to rigorously assess the quality of LLM judges. We manually select approximately 25% of the U-MATH problem statements and golden answers, supplying each with four solutions produced by different top-performing LLMs, and label them based on whether the generated solutions are correct or not. The benchmark is designed to be challenging for LLM judges yet representative of the typical university-level math grading tasks.

3. **Comparison of Models** (Section 4): We conduct a comparative analysis of various open-source and proprietary LLMs on U-MATH. Our analysis highlights the high performance of specialized models in text-only problems and the superiority of proprietary models in visual tasks with the best U-MATH accuracy of 49%. Additionally, we examine several popular LLMs on $\mu$-MATH to assess their ability to judge free-form mathematical problems. Our results show the best model achieving the macro F1-score of 80%.

We release the U-MATH and $\mu$-MATH benchmarks under a permissive license to facilitate further research and ensure reproducibility.

## 2 BACKGROUND

Enhancing and evaluating the mathematical reasoning capabilities of LLMs is essential in AI research (Ahn et al., 2024). Studies show that finetuning with mathematical and code-related data enhances models' general skills (Prakash et al., 2024). Mathematical tasks require logical thinking and multi-step problem-solving, thus improving overall reasoning abilities in LLMs (Chen et al., 2024).

This leads to the problem of evaluating LLM's math abilities. Despite the significant progress, many existing benchmarks are limited in scope, focusing primarily on school-level mathematics or limited in size and topic coverage. Table 1 summarizes popular text-only and visual mathematical benchmarks.

| Dataset | Levels | %Uni. Level | #Test | %Visual | %Free Form Answer |
|---|---|---|---|---|---|
| MMLU$_{Math}$ (Hendrycks et al., 2020) | E H C | 0 | 1.3k | 0 | 0 |
| GSM8k (Cobbe et al., 2021) | E | 0 | 1k | 0 | 0 |
| MATH (Hendrycks et al., 2021) | H O | 0 | 5k | 0 | 100 |
| MiniF2F (Zheng et al., 2021) | E H O | 0 | 244 | 0 | 100 |
| OCWCourses (Lewkowycz et al., 2022) | U | 100 | 272 | 0 | 100 |
| ProofNet (Azerbayev et al., 2023) | C U | ≈50 | 371 | 0 | 100 |
| CHAMP (Mao et al., 2024) | H | 0 | 270 | 0 | 100 |
| MathOdyssey (Fang et al., 2024) | H U O | 26 | 387 | 0 | 100 |
| MMMU$_{Math}$ (Yue et al., 2023) | C | 0 | 505 | 100 | 0 |
| MathVista (Lu et al., 2023) | E H C | 0 | 5k | 100 | 46 |
| MATH-V (Wang et al., 2024a) | E H O | 0 | 3k | 100 | 50 |
| We-Math (Qiao et al., 2024) | E H U | ≈20 | 1.7k | 100 | 0 |
| MathVerse (Zhang et al., 2024) | H | 0 | 4.7k | 83.3 | 45 |
| **U-MATH** (this work) | U | 100 | 1.1k | 20 | 100 |

Table 1: Existing Auto-evaluation Math benchmarks with corresponding test samples *published*, visual samples percent, and percent of multiple-choice questions. Level denotes E Elementary to Middle School, H High School, C College, U University, O Different Olympiads.

**Textual Mathematical Benchmarks.** Early efforts to assess LLMs' mathematical abilities have emerged in datasets like MathQA (Amini et al., 2019) and the mathematics subset of MMLU (Hendrycks et al., 2020). These early benchmarks emphasized the importance of operation-based reasoning in solving mathematical word problems, typically in a multiple-choice format. Nowadays, even smaller models (e.g., 7B parameters) have achieved high scores on these tasks (Li et al., 2024b), suggesting that these benchmarks are becoming saturated. In response, more comprehensive datasets have emerged, such as GSM8K (Cobbe et al., 2021) and MATH (Hendrycks et al., 2021), or MGSM (Shi et al., 2022) (multilingual version of 250 GSM8K samples). These popular benchmarks are crucial for evaluating LLMs' mathematical reasoning skills. However, they primarily focus on school-level problems, which may not fully assess the depth of mathematical reasoning.

Recent efforts attempt to address more advanced mathematical concepts. MathOdyssey (Fang et al., 2024) with competition problems, OCWCourses (Lewkowycz et al., 2022) from actual MIT courses, and ProofNet (Azerbayev et al., 2023) focusing on proofs aim to evaluate undergraduate-level or olympiad-level knowledge. However, these datasets are constrained by their small sizes (e.g., 387, 272, and 371 samples), limiting their statistical robustness and topic coverage. For example, MathOdyssey is limited to 101 samples in university-level topics (Calculus, Algebra, and Diff. Equations and Statistics). Other specialized datasets like MiniF2F (Zheng et al., 2021) provide valuable parallel corpora in formal languages, while CHAMP (Mao et al., 2024) offers helpful context and hints, but both are similarly limited in scale with 244 and 270 samples. Additionally, both heavily rely on already published resources: CHAMP sources material from a book, while MiniF2F re-uses international olympiads and MATH dataset problems. An attempt to provide a more robust evaluation, GHOSTS (Frieder et al., 2024) dataset, provides 728 problems (both from other datasets and new ones) but does not provide reference solutions and answers, focusing instead on human evaluation, making cheap automatic evaluation impossible.

The current datasets are either too small, leading to higher measurement errors, or focus mainly on elementary and high school math, leaving a gap in evaluating LLMs' proficiency in advanced university-level math topics.

**Visual Mathematical Benchmarks.** As multimodal LLMs gain prominence, there is a growing need for visual mathematical benchmarks (Zhang et al., 2024; Qiao et al., 2024). Early efforts in this domain focus primarily on geometric problems, as seen in datasets like GeoQA (Chen et al., 2022b), UniGeo (Chen et al., 2022a), and Geometry3K (Lu et al., 2021). These datasets have a narrow focus that does not encompass the breadth of mathematical visual reasoning required at advanced levels.

More recent benchmarks attempt to broaden the scope of visual mathematical evaluation. One of the first comprehensive attempts is the mathematical subset of MMMU (Yue et al., 2023), which offers 505 college-level multiple-choice questions, all with images. However, its multiple-choice format limits the complexity of problems that can be posed. MathVista (Lu et al., 2023) collects 28 existing datasets and introduces 3 new datasets with a total of 5k samples (1k testmini samples). However, as shown by Qiao et al. (2024), it faces challenges with data quality due to its compilation from older datasets.

The latest benchmarks, such as MATH-V (Vision) (Wang et al., 2024a) and We-Math (Qiao et al., 2024), extend this approach to collect 3k and 1.7k visual samples, respectively. However, both datasets rely on multiple-choice questions in the test set, leading to faster saturation. MathVerse (Zhang et al., 2024) further extends this approach, relying on visual elements and providing some simple text problems with 1.2k brand-new samples. Among these, only the We-Math dataset includes university-level mathematical problems.

Our U-MATH dataset improves on existing benchmarks with 225 of 1,100 university-level problems that require visual elements (graph, table, diagram) to be solved. This balanced ratio ensures models are challenged to handle both traditional and visual problem-solving without over-relying on visuals, mirroring real-world scenarios.

**Large Language Models for Mathematics.** The application of LLMs to mathematical problem-solving shows promising results, particularly with models like GPT-3.5 and GPT-4 demonstrating strong reasoning abilities for complex tasks such as those in the MATH dataset (Achiam et al., 2023). While open-source models initially lagged in performance on advanced mathematical tasks, the Llama-3.1 (Dubey et al., 2024) is approaching parity with proprietary models. The most popular benchmarks, MATH and GSM8K, are nearing saturation, with Llama 3.1 405B achieving scores of 73.8% and 96.8%, respectively. Similarly, a Qwen2.5-Math-72B model (Yang et al., 2024b; Team, 2024) reach 85.9% on MATH while Qwen2-Math-72B (Yang et al., 2024a) reaches 96.7% on GSM8k.

To enhance LLMs' mathematical capabilities, researchers develop various prompt-based methods (Liu et al., 2021). These include techniques for encouraging chain-of-thought generation (Wei et al., 2022), selecting final results from multiple sampled outputs (Wang et al., 2022), and using external tools such as calculators, WolframAlpha or Python interpreters (Gao et al., 2023) to reduce arithmetic errors. Additionally, instruction tuning during pre-training has been identified as a key factor in improving performance (Wang et al., 2017). While these approaches show promise, their effectiveness on university-level problems still needs to be explored due to the lack of suitable large-scale benchmarks.

**Mathematical solution verification.** Evaluating mathematical solutions is uniquely challenging due to the open-ended nature of answers and the inherent ambiguity in mathematical expressions. Consequently, many benchmarks opt for multiple-choice formats due to their grading simplicity. However, this approach often simplifies tasks, providing hints that models can exploit (Li et al., 2024c; Pezeshkpour and Hruschka, 2023).

While free-form evaluation using LLM judges is widespread (Zheng et al., 2023), it is known to introduce potential errors (Zheng et al., 2023), since evaluating mathematical solutions is a complex task in its own right (Zeng et al., 2023; Xia et al., 2024). These evaluation errors are largely overlooked and unaccounted for, limiting the reliability of inferences drawn from such evaluations.

Hence, it is important to be able to estimate the performance of automatic evaluators and to choose the most adequate among them. Recent studies show that evaluation performance is correlated with but does not equal problem-solving performance (Stephan et al., 2024). This underscores the importance of benchmarks designed specifically to asses the evaluators — also called meta-evaluations.

There are existing benchmarks that are well-suited for meta-evaluations. PRM800K (Lightman et al., 2023) contains 800K annotated steps from 75K solutions to 12K MATH dataset problems, designed

to confuse reward models. FELM (Zhao et al., 2024) provides GPT-3.5 annotations for solutions to 208 GSM8K and 194 MATH problems. MR-GSM8K (Zeng et al., 2023) and MR-MATH (Xia et al., 2024) introduce meta-evaluation datasets focused on the GSM8K and MATH datasets, respectively. However, these are either based on elementary to high-school level problems or feature specifically competition-style math, leaving a gap in meta-evaluations on complex and practical university tasks.

To address this, we introduce $\mu$-MATH— a meta-evaluation dataset based on a subset of U-MATH problems. It provides LLM-generated solutions with verified labels, enabling precise and fine-grained assessment of LLMs' evaluation abilities.

## 3 U-MATH

We present **U-MATH** (stands for University Math) — a benchmark designed to challenge LLMs with problems requiring deep understanding and advanced reasoning. The problems span 6 core topics and range in difficulty and number of questions. A subset of 20% of problems includes images to test the models' ability to interpret and reason with graphical information. Reference solutions and answers accompany all problems.

**Accuracy** is the primary performance metric for **U-MATH**, its text-only problems (**U-MATH$_T$**) and problems that include a visual component (**U-MATH$_V$**). The main performance measure for $\mu$-**MATH** is **macro-F1**.

We use an LLM as a judge (Zheng et al., 2023) to measure the accuracy of the free-form answers against the golden solutions. A problem is considered solved only if all required questions are answered and all requested items (e.g., all saddle points) are correctly identified.

### 3.1 DATASET COLLECTION

To create a benchmark that authentically reflects university-level mathematics, we collaborate with [ANONYMIZED], a platform providing learning content and software for top US universities specialized in mathematics. The problems are sourced from ongoing courses across various institutions currently run on the [ANONYMIZED] platform. Problems and solutions are crafted by subject matter experts and represent real-world academic standards. These samples are unpublished and have not been exposed to any external sources. Thus, the dataset could not be leaked to current LLMs.

We employ a multi-stage filtering process to select challenging problems from tens of thousands of available samples. First, we filter out problems with short solutions ($< 100$ characters) and problems in multiple-choice format. As LLMs are not designed to perform arithmetic calculations and are prone to errors (Hendrycks et al., 2021; Lewkowycz et al., 2022), we focus on testing mathematical reasoning rather than calculations. We filter out problems marked as allowing calculator usage. As for the visual problems selection, we chose to keep problems with a single image for convenience.

Next, we employ several small LLMs (LLaMA-3.1-8B (Dubey et al., 2024), Qwen2-7B (Yang et al., 2024a), Mistral-7B (Jiang et al., 2023), Mathstral-7B, NuminaMath-7B (Beeching et al., 2024)) to solve the problems. We select 150 most challenging problems for each subject based on the average problem solution rate. For this step, we use the same pipeline as described in Section 4. This way, we ensure that none of the individual models influence problem selection largely and that there is no overfitting to a specific LLM. As the last step, we hold extra validation high risk problems (with low solve rate) using our in-house math experts and [ANONYMIZED] content team.

After collection, we enlist a team of paid experts from the [ANONYMIZED], who actively teach various Calculus courses. The experts verify that each problem is suitable either for assessing the subject knowledge expected of college or university students or for testing prerequisite knowledge. The team thoroughly reviewed and affirmed that the selected problems meet these criteria. Overall, only 4.3% of the problems are categorized as school-level rather than university-level, highlighting the robustness of the selection process.

## 3.2 DATASET STATISTICS

The U-MATH benchmark comprises **1,100** carefully curated and validated mathematical problems. These problems are distributed across **6 core subjects** with about 20% of the tasks incorporating visual elements, such as graphs, tables, and geometric figures, mirroring the multi-modal nature of real-world mathematical problems: Precalculus (Review), Algebra, Differential Calculus (+Differential Equations), Integral Calculus, Multivariable Calculus, and Sequences & Series.

| Math Subject | #Textual | #Visual | Avg. Questions | Avg. Answers |
|---|---|---|---|---|
| Algebra | 150 | 30 | 1.93 | 1.28 |
| Differential Calculus | 150 | 70 | 2.37 | 1.15 |
| Integral Calculus | 150 | 58 | 1.09 | 1.01 |
| Multivariable Calculus | 150 | 28 | 1.74 | 1.09 |
| Precalculus | 150 | 10 | 1.51 | 1.23 |
| Sequences and Series | 150 | 4 | 1.36 | 1.00 |
| All | 900 | 200 | 1.66 | 1.12 |

Table 2: Average number of questions per problem and answers per question in U-MATH.

Table 2 summarizes the distribution of problems across different subjects. The average is **1.7** questions per problem (e.g., local minima, maxima, and increasing intervals are asked), and the average of **1.1** answers per question (for example, the number of saddle points in the correct answer).

## 3.3 META-EVALUATION FRAMEWORK ($\mu$-MATH)

Mathematical problem evaluation is not straightforward. Even simple expressions like $x \cdot 0.5$ may have valid forms like $\frac{x}{2}$, $x \div 2$, $x/2$, or unsimplified variants like $9x/18$. In practice, evaluating free-form solutions requires testing expression equivalence in much less trivial cases, especially with more advanced problems (refer to Section A.3 in Appendix for an example).

To systematically study the ability of LLMs to evaluate free-form mathematical solutions on advanced, university-level problems, we introduce the **$\mu$-MATH** (Meta U-MATH) benchmark. It consists of a curated subset of U-MATH samples, supplied with LLM-generated solutions — both correct and not. The solutions are labeled using a combination of manual inspection and automated verification via [ANONYMIZED]-API, which allows to test formal equivalence of mathematical expressions.

We selected 271 U-MATH problems (around **25%**) based on their assessment difficulty to create a challenging meta-evaluation set. This subset does not aim to reflect the overall U-MATH distribution but rather to provide a robust test for LLM judges. We focused on text-only problems, excluding those needing images, due to the limited size of the labeled U-MATH subset. Four solutions have been generated for each of the selected problems — using Qwen2.5-72B, Llama3.1-8B, GPT-4o and Gemini-1.5-Pro models — **1084 samples** in total.

A tested model is provided with a problem statement, a reference answer, and a solution to evaluate. We treat this as a binary classification task, using the macro-averaged **F1-score as the primary metric** to minimize the effect of class imbalance. Additionally, we report Positive Predictive Value (PPV or Precision) and True Positive Rate (TPR or Recall) for the positive class as well as Negative Predictive Value (NPV) and True Negative Rate (TNR) for the negative class, offering a finer-grained performance evaluation. We also report all of the scores computed both across the entire set of samples and only across those with solutions produced by a specific model, separately for each of the author models.

## 4 EXPERIMENTS AND RESULTS

### 4.1 EXPERIMENTAL SETUP

We select some top-performing recent LLMs to evaluate.

| Model | Source | Size(s) | Visual | Open-weights |
|---|---|---|---|---|
| Mathstral-v0.1 | (Mistral.ai, 2024) | 7B | ✗ | ✓ |
| NuminaMath-CoT | (Beeching et al., 2024) | 7B | ✗ | ✓ |
| LLaMA-3.1 | (Dubey et al., 2024) | 8B, 70B | ✗ | ✓ |
| LLaMA-3.1-Nemotron | (Wang et al., 2024b) | 70B | ✗ | ✓ |
| Qwen2-Math | (Yang et al., 2024a) | 7B, 72B | ✗ | ✓ |
| Qwen2.5-Math | (Yang et al., 2024b) | 7B, 72B | ✗ | ✓ |
| Qwen2.5 | (Team, 2024) | 7B, 72B | ✗ | ✓ |
| Athene-V2-Chat | (Nexusflow, 2024) | 72B | ✗ | ✓ |
| Pixtral-12B-2409 | (Mistral AI, 2024) | 12B | ✓ | ✓ |
| LLAVA One Vision $_{\text{Qwen2-7B}}$ | (Li et al., 2024a) | 8B | ✓ | ✓ |
| Qwen2-VL | (Yang et al., 2024a) | 7B, 72B | ✓ | ✓ |
| LLaMA-3.2 | (Meta AI, 2024) | 11B, 90B | ✓ | ✓ |
| Claude-3.5-Sonnet | (Anthropic, 2024) | unknown | ✓ | ✗ |
| GPT-4o-mini-2024-07-18 | (OpenAI, 2024) | unknown | ✓ | ✗ |
| GPT-4o-2024-08-06 | (OpenAI, 2024) | unknown | ✓ | ✗ |
| Gemini-1.5-Flash-002 | (Team et al., 2024) | unknown | ✓ | ✗ |
| Gemini-1.5-Pro-002 | (Team et al., 2024) | unknown | ✓ | ✗ |

Table 3: LLMs name, version and sizes we evaluate.

All LLMs are tested using the same prompts and settings for fair comparison. The LLMs are restricted to a single generation of 4096 tokens with the temperature set to 0. We employ chain-of-thought (CoT) prompting (Wei et al., 2022) to encourage models to 'think' before providing an answer. Images are included directly in the prompts for multimodal LLMs. To text-only LLMs the problem description is provided as-is without visual elements.

We report accuracy based on widely available GPT-4o-2024-08-06 as-a-judge for the final results, despite this not being the best judge, yet conservative in false negative rate (as discussed in Section 4.3). The judge is presented with the problem statement, golden answer, and generated solutions. The temperature is set to 0. The judge is asked to extract the 'student's answer', make derivations that may be necessary, and compare solutions. After this 'reasoning phase', we ask the model to provide a Yes/No response given previous reasoning, which we interpret as a desired binary metric.

For meta-evaluation we additionally experiment with two prompting schemes — a standard Automatic CoT prompt with a simple task description and an instruction to think step-by-step, and a manual CoT prompt with explicit instructions on how to tackle the task. We find the best performance to be achieved with the latter, so we use the manual prompt (referenced as CoT) for the main results. The CoT output is then given to the extractor model (we always use Qwen2.5 72B for consistency) to produce a single label — either 'Yes', 'No' or 'Inconclusive'. We include 'Inconclusive' for cases when a judge refuses to complete the evaluation or generation fails; such judgments are treated as incorrect. Refer to Appendix C.2 for full prompts.

## 4.2 U-MATH RESULTS

Figure 2 compare popular text-only and multimodal models in U-MATH as well as U-MATH$_{\text{Text}}$ and U-MATH$_{\text{Visual}}$. Table 4 summarizes the performance of all evaluated LLMs on the U-MATH benchmark. Reference to Appendix E for model performance vs model size comparison.

Among text-only models, the math-specific model Qwen2.5-Math-72B achieves the highest overall accuracy at 50.2%, showcasing strong mathematical reasoning capabilities. In the multi-modal model group, **Gemini-1.5-pro-002 leads** with an overall accuracy of **60.1%**, highlighting the advantages of integrating visual processing. In contrast, best open-weights model Qwen2-VL-72B lacks mathematical abilities in visual and textual tasks with 31.2% on a U-MATH benchmark. Building on these results, several key trends emerge:

- **Model Size vs. Specialization:** Larger models expectedly outperform smaller ones. However, the small specialized model Qwen2.5-Math-7B surpasses or performs on par with 10 times larger models like Qwen2.5-72B or LLaMA-3.1-70B and almost reaching leading Gemeni-1.5-Pro level.

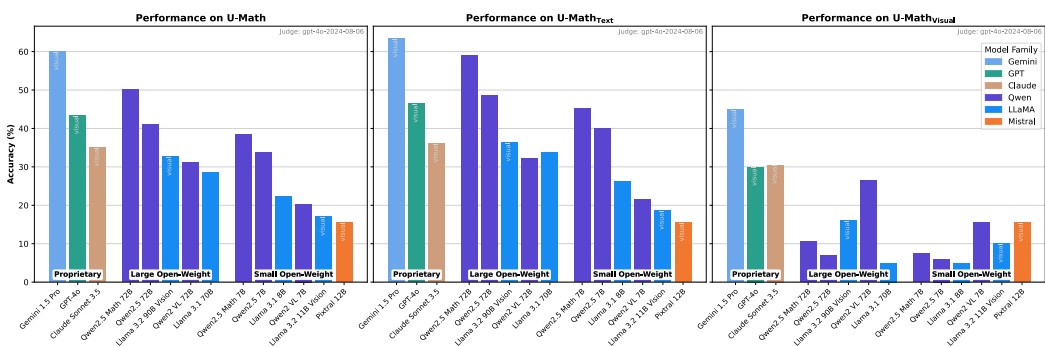

Figure 2: Performance of the selected top-performing models on U-MATH, U-MATH_Text and U-MATH_Visual. Color denotes different model families, 'visual' label highlight visual encoder of the model. Higher is better for all charts.

| Model | U-MATH | U-MATH T 900 | U-MATH V 200 | Algebra T 150 | Algebra V 30 | Diff. C. T 150 | Diff. C. V 70 | Integral C. T 150 | Integral C. V 58 | Multivar C. T 150 | Multivar C. V* 28 | Precalculus T 150 | Precalculus V* 10 | Seq.& Series T 150 | Seq.& Series V* 4 |
|---|---|---|---|---|---|---|---|---|---|---|---|---|---|---|---|
| *Text-only models* | | | | | | | | | | | | | | | |
| Mathstral 7B | 18.0 | 20.7 | 6.0 | 51.3 | 6.7 | 4.0 | 10.0 | 1.3 | 1.7 | 8.0 | 3.6 | 48.7 | **10.0** | 10.7 | 0.0 |
| NuminaMath 7B | 19.2 | 22.8 | 3.0 | 62.7 | 0.0 | 4.0 | 7.1 | 1.3 | 0.0 | 6.0 | 3.6 | 51.3 | 0.0 | 11.3 | 0.0 |
| Llama-3.1 8B | 22.3 | 26.1 | 5.0 | 59.3 | 3.3 | 6.7 | 5.7 | 9.3 | 3.4 | 11.3 | 3.6 | 54.7 | **10.0** | 15.3 | 25.0 |
| Qwen2.5 7B | 33.8 | 40.0 | 6.0 | 86.0 | **10.0** | 12.7 | 1.4 | 10.0 | 12.1 | 26.7 | 3.6 | 75.3 | 0.0 | 29.3 | 0.0 |
| Qwen2.5-Math 7B | 38.4 | 45.2 | 7.5 | 87.3 | 6.7 | 18.7 | 5.7 | 8.0 | 10.3 | 36.0 | 10.7 | 80.7 | 0.0 | 40.7 | 0.0 |
| NuminaMath 72B | 25.0 | 29.7 | 4.0 | 74.7 | 3.3 | 6.7 | 4.3 | 4.0 | 3.4 | 11.3 | 3.6 | 62.7 | 10.0 | 18.7 | 0.0 |
| Llama-3.1 70B | 28.5 | 33.7 | 5.0 | 82.0 | 3.3 | 10.7 | 5.7 | 4.0 | 5.2 | 14.0 | 3.6 | 64.0 | 0.0 | 27.3 | 25.0 |
| Llama-3.1 Nemotron 70B | 31.4 | 37.4 | 4.0 | 84.0 | 0.0 | 14.7 | 2.9 | 4.0 | 3.4 | 25.3 | 7.1 | 64.0 | 20.0 | 32.7 | 0.0 |
| Qwen2.5 72B | 41.0 | 48.6 | 7.0 | 88.7 | 6.7 | 22.7 | 4.3 | 12.0 | 6.9 | 40.0 | 17.9 | 83.3 | 0.0 | 44.7 | 0.0 |
| Athene-V2 72B Chat | 46.2 | 54.6 | 8.5 | 88.7 | 3.3 | 34.0 | 4.3 | 16.0 | 6.9 | 50.7 | **21.4** | 88.7 | **10.0** | 49.3 | **50.0** |
| Qwen2.5-Math 72B | **50.2** | **59.0** | **10.5** | **92.7** | 6.7 | **35.3** | 7.1 | **20.7** | **17.2** | **58.0** | 7.1 | **90.0** | 0.0 | **57.3** | **50.0** |
| *Multimodal models* | | | | | | | | | | | | | | | |
| Pixtral 12B | 15.5 | 15.6 | 15.5 | 44.7 | 23.3 | 1.3 | 34.3 | 0.7 | 0.0 | 3.3 | 0.0 | 32.0 | 0.0 | 11.3 | 0.0 |
| Llama-3.2 11B Vision | 17.0 | 18.6 | 10.0 | 54.0 | 10.0 | 1.3 | 20.0 | 1.3 | 1.7 | 4.7 | 3.6 | 43.3 | 10.0 | 6.7 | 0.0 |
| LLaVA-OV Qwen2-7B | 17.7 | 20.7 | 4.5 | 60.7 | 6.7 | 4.0 | 5.7 | 1.3 | 1.7 | 5.3 | 3.6 | 43.3 | 10.0 | 9.3 | 0.0 |
| Qwen2-VL 7B | 20.4 | 21.4 | 15.5 | 62.7 | 10.0 | 4.7 | 32.9 | 0.7 | 5.2 | 6.7 | 7.1 | 45.3 | 0.0 | 8.7 | 0.0 |
| Qwen2-VL 72B | 31.2 | 32.2 | 26.5 | 80.7 | 26.7 | 9.3 | 40.0 | 2.0 | 13.8 | 14.7 | 28.6 | 65.3 | 10.0 | 21.3 | 0.0 |
| Llama-3.2 90B Vision | 32.6 | 36.3 | 16.0 | 85.3 | 26.7 | 10.7 | 25.7 | 2.7 | 1.7 | 22.7 | 7.1 | 65.3 | 20.0 | 31.3 | 25.0 |
| Claude Sonnet 3.5 | 35.1 | 36.1 | 30.5 | 76.0 | 33.3 | 12.0 | 41.4 | 7.3 | 17.2 | 21.3 | 28.6 | 65.3 | 30.0 | 34.7 | 25.0 |
| GPT-4o-mini | 37.2 | 40.3 | 23.0 | 88.0 | 16.7 | 16.7 | 31.4 | 4.0 | 10.3 | 24.0 | 35.7 | 77.3 | 20.0 | 32.0 | 25.0 |
| GPT-4o | 43.5 | 46.4 | 30.0 | **91.3** | 30.0 | 18.7 | 32.9 | 10.0 | 20.7 | 41.3 | 42.9 | 79.3 | 30.0 | 38.0 | 25.0 |
| Gemini 1.5 Flash | 51.3 | 53.8 | 40.0 | **91.3** | 50.0 | 36.0 | 45.7 | 14.0 | **24.1** | 44.0 | 50.0 | 80.7 | 30.0 | 56.7 | **50.0** |
| Gemini 1.5 Pro | **60.1** | **63.4** | **45.0** | **91.3** | **60.0** | **50.7** | **47.1** | **27.3** | **24.1** | **60.7** | **57.1** | **87.3** | **70.0** | **63.3** | **50.0** |

Table 4: Comparison of models' accuracy on our U-MATH benchmark and its subjects. Scores for various mathematical categories, including text and visual analysis, are displayed. For each subject 2 numbers are provided - text-only (T) and visual (V) problems. Asterisk denotes a small number of samples ($< 30$). Free-form solutions judged by gpt-4o-2024-08-06. Images are not included in the prompt for text-only models, only the problem statement. **Bold** indicates the best result in each group.

On the other hand, Pixtral-12B performs consistently worse than minor Qwen2-VL-7B, indicating a lack of university-level data in training.

- **Textual vs. Visual Problem-Solving:** Across multimodal models, text-only problems' accuracy vastly exceeds visual problems, highlighting areas for further improvement. The text-only models can solve a small percentage of visual problems, primarily due to guessing or judging errors discussed in Section 4.3.

- **Proprietary vs. Open-weights model:** Proprietary models like Gemini still offer top or competitive performance but lack transparency and flexibility. At the moment, the gap is evident in visual comprehension, with 18.5% difference on U-MATH_Visual between top-1 and best open-weight model. However, open-weight models like Qwen-Math is a big step toward top performance.

- **Continuous Finetuning:** Additional tuning significantly enhances performance, with LLaMA-3.1 70B ⇒ LLaMA-3.1 Nemotron 70B and Qwen2.5-72B ⇒ Athene-V2 72B achieving 2.9% and 5.2% higher U-MATH accuracy, respectively. This reinforces the idea that models are not fully optimized for their size and require high-quality data for further improvements.

**Subject-Specific Results** Model performance varies across mathematical subjects, excelling in text-based tasks for Precalculus and Algebra, consistent with benchmark saturation (Ahn et al., 2024), but faltering on visual-symbolic tasks. In Sequences and Series, success on formula-based problems reflects logical structuring, though limited visual data restricts evaluation. Differential and Multivariable Calculus results are moderate, with difficulties in abstract, multi-dimensional concepts, especially visual interpretations. Integral Calculus presents the greatest challenge, as interpreting curves, areas, and extensive expressions confounds models, underscoring the need for improved multimodal training.

## 4.3 META-EVALUATION ($\mu$-MATH) RESULTS

| Model | U-MATH$_{\text{Text}}$ | $\mu$-MATH | | | | | $\mu$-MATH$_{\text{Qwen}}$ F1 | $\mu$-MATH$_{\text{Llama}}$ F1 | $\mu$-MATH$_{\text{GPT}}$ F1 | $\mu$-MATH$_{\text{Gemini}}$ F1 |
| | | F1 | TPR | TNR | PPV | NPV | | | | |
|---|---|---|---|---|---|---|---|---|---|---|
| Llama-3.1 8B | 26.1 | 52.0 | 48.7 | 55.9 | 56.0 | 48.5 | 48.7 | 49.2 | 51.2 | 55.5 |
| Qwen2.5 7B | 40.0 | 69.3 | **78.7** | 59.8 | 69.3 | **70.8** | 62.4 | 72.3 | 68.3 | 69.1 |
| Qwen2.5-Math 7B | 45.2 | 61.9 | 76.6 | 47.9 | 62.9 | 63.9 | 59.7 | 63.8 | 57.2 | 63.8 |
| GPT-4o-mini | 40.3 | 72.3 | 59.0 | 88.1 | 85.1 | 65.1 | 69.3 | 76.2 | **70.4** | 69.6 |
| Gemini 1.5 Flash | **53.8** | 74.8 | 63.3 | **88.3** | **86.2** | 67.6 | **71.2** | **80.6** | 70.1 | **73.9** |
| LLaMA-3.1-70B | 33.7 | 61.0 | 62.5 | 59.6 | 64.1 | 57.9 | 56.0 | 57.0 | 69.4 | 58.8 |
| Qwen2.5 72B | 48.6 | 75.6 | 77.1 | 74.2 | 77.5 | 73.7 | 70.5 | 79.3 | 73.7 | 74.2 |
| Qwen2.5-Math 72B | 59.0 | 74.0 | **80.9** | 66.8 | 73.8 | 75.2 | 69.3 | 77.3 | 68.2 | 76.8 |
| Claude 3.5 Sonnet | 36.1 | 74.8 | 62.5 | **89.5** | 87.3 | 67.4 | 70.8 | 77.9 | 72.2 | 73.8 |
| GPT-4o | 46.4 | 77.4 | 70.1 | 85.9 | 85.1 | 71.3 | 74.2 | 81.8 | 77.5 | 72.6 |
| Gemini 1.5 Pro | **63.4** | **80.7** | 77.5 | 84.5 | 85.2 | **76.4** | **77.7** | **83.6** | **78.2** | **79.5** |

Table 5: Comparison of model's ability to judge on $\mu$-MATH benchmark using CoT prompt; Macro F1-score (F1), True Positive Rate (TPR), True Negative Rate (TNR), Positive Predictive Value (PPV), and Negative Predictive Value (NPV), with F1 as the primary one are presented. Columns under $\mu$-MATH represent the integral score over the entire benchmark, while $\mu$-MATH $_{\text{<model>}}$ is a subset with solutions generated by a specific model. U-MATH$_{\text{Text}}$ accuracy is added for comparison of model's performance as a math solver vs as a math judge. **Bold** indicates the best result within a column. Reference full table in Appendix J.

We find that **using manual CoT instructions instead of Automatic CoT improves or maintains judgment performance**, for all but the LLaMA models, as shown in Table 5. LLaMA's performance drop stems from higher inconclusive judgment rates with CoT (refer to Appendix G). At the same time, Gemini models benefit the most from this transition, gaining over 10% in F1-score and becoming the top-ranked models, surpassing Qwen and GPT models that outperform Gemini with Automatic CoT. Appendix F provides this data and a visual comparison.

It is also evident that **being a better solver does not necessarily lead to being a better judge**, see additional discussion in Appendix H. Also, the best attainable **overall F1-score is only 80.7%**, which constitutes a significant gap in the context of judgment. Error rates of the judges directly limit the precision of capability evaluations, potentially even biasing them due to the possibility of errors being systematic in nature as opposed to pure noise.

The results reveal a consistent **bias towards some models** (better performance on LLaMA solutions and worse performance on Qwen solutions), most pronounced with smaller models and Automatic CoT prompt. This bias is reduced for both small and large models when transitioning to CoT prompting, which is also illustrated with Figure 3. At the same time, no noticeable 'self-judgment' bias is found.

Besides that, we observe a substantive difference in judges' behavior: **proprietary models tend to be more conservative** — having relatively high TPR compared to their TNR, while **Qwen family of models exhibits the opposite pattern**. Furthermote, proprietary models mainly lose in TPR when going from a larger model to a smaller one, while Qwen models, once again on the contrary, lose more in TNR. The behavior differences are further studied in Appendix I.

Overall, judges have imperfect performance, they exhibit varying behavior patterns, judgment performance is different from problem-solving performance, and different prompting schemes induce nontrivial changes in judges' behaviors, biases and even their rankings. All of these findings underscore the importance of performing meta-evaluations, since such things are impossible to quantify and comparisons impossible to make in the absence of benchmarks designed for judges.

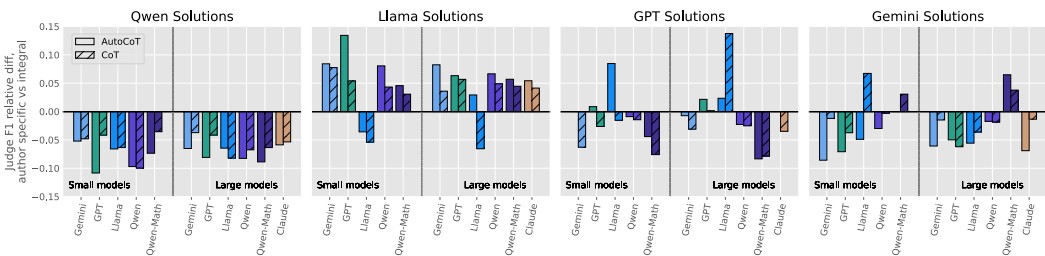

Figure 3: Relative differences between specific judgment performance — over samples with solutions generated by a specific author model — and integral judgment performance — across all the samples. The judgment performance is measured by the $\mu$-MATH macro F1-scores. Each pane corresponds to a different author model used when measuring specific performance. The x-axis specifies which judge corresponds to a particular bar pair, with bar pairs comparing the relative diffs in case of Automatic CoT and CoT prompting schemes.

## 5 CONCLUSION

We introduce **U-MATH**, a novel multimodal benchmark for evaluating the university-level mathematical reasoning of LLMs. U-MATH includes 1,100 unpublished free-form problems from real teaching materials, covering 6 core mathematical subjects, with 20% involving image-based reasoning. Additionally, we provide $\mu$-**MATH**, a meta-evaluation dataset, to assesses LLMs' ability to evaluate free-form mathematical solutions.

Our experiments highlight significant challenges for LLMs in advanced reasoning and visual problem-solving. The highest accuracy achieved was 63.4% on text-based tasks and 45.0% on visual problems (Gemini-1.5-pro-002). Solution assessment remains difficult, with Gemini hiy top $\mu$-MATH F1-score of 80%, showing room for improvement and underscoring the limitations of widely used models like GPT-4o in evaluation tasks.

**Limitations.** While U-MATH offers diverse university-level problems, it does not cover the full range of advanced topics and may introduce biases by favoring certain problem types. Also, selection process may introduce biases, potentially favoring certain problem types or difficulty levels (e.g., more accessible topics like Precalculus and Algebra). The inclusion of 20% visual problems, yet reflect real distribution, limits the evaluation of visual reasoning. Furthermore, reliance on LLMs for valuation introduces potential, as models struggle with complex reasoning and instructions, evidenced by our findings with the $\mu$-MATH. The $\mu$-MATH dataset encompass of 25% of U-MATH problems narrows the evaluation scope, but provide 4 diverse model families as solution generators.

**Future Work.** Future research can focus on enhancing LLM performance by integrating existing tool-augmented models and exploring their effectiveness on U-MATH and $\mu$-MATH tasks. For instance, incorporating external tools, such as formal solvers, could improve complex textual and multimodal reasoning capabilities. Additionally, our findings indicate that widely used models like GPT-4o are not a silver bullet for solution evaluation; thus, developing specialized (finetuned) models or techniques for more accurate and unbiased assessment is a promising direction. Expanding $\mu$-MATH with formal verification methods could further enhance the evaluation processes. Additionally, conducting deeper prompt sensitivity analyses would provide valuable insights for the field.

By open-sourcing U-MATH, $\mu$-MATH, and the evaluation code, we aim to facilitate further research in advancing the mathematical reasoning capabilities of LLMs and encourage the development of models better equipped to tackle complex, real-world mathematical problems.

## ETHICS STATEMENT

We collected all data in U-MATH and $\mu$-MATH with appropriate permissions, ensuring no personal or proprietary information is included. The datasets consist solely of mathematical problems and solutions, without any sensitive content. The annotators from [ANONYMIZED] are employed in

the partner laboratory with [ANONYMIZED]; their annotation time is fully compensated at a fair hourly rate. We open-sourced the datasets and code under suitable licenses to support transparency and research advancement. There are no known conflicts of interest associated with this work.

## REPRODUCIBILITY STATEMENT

All datasets and code will be available on GitHub. Detailed descriptions of dataset collection and processing are in Section 3. The experimental setup, including model configurations and prompts, is outlined in Section 4, with full prompts provided in Appendices C.1 and C.2. These resources enable replication of our experiments.

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

# A  PROBLEM EXAMPLES

## A.1  U-MATH PROBLEMS

---

**Example 1: Algebra.**

Write a logarithmic equation corresponding to the graph shown. Use $\log_3(x)$ as a parent function:

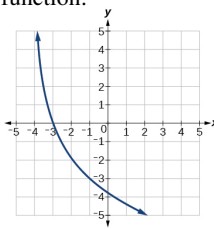

---

The final answer: $-3 \cdot \log_3(x + 4)$

---

**Example 2: Integral Calculus.**

Solve the integral:

$$\int \frac{-9 \cdot \sqrt[3]{x}}{9 \cdot \sqrt[3]{x^2} + 3 \cdot \sqrt{x}} \, dx$$

---

$$-\frac{2}{27} \cdot \ln\left(\frac{1}{3} \cdot \left|1 + 3 \cdot \sqrt[6]{x}\right|\right)$$
$$-\frac{1}{3} \cdot \sqrt[6]{x}^2 - \frac{3}{2} \cdot \sqrt[6]{x}^4 + \frac{2}{3} \cdot \sqrt[6]{x}^3$$
$$+\frac{2}{9} \cdot \sqrt[6]{x} + C$$

---

**Example 3: Precalculus Review.**

Find a formula for $f(x)$, the sinusoidal function whose graph is shown below:

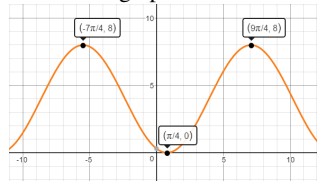

---

The final answer:
$f(x) = -4 \cdot \cos\left(\frac{1}{2} \cdot \left(x - \frac{\pi}{4}\right)\right) + 4$

---

**Example 4: Multivariable Calculus.**

$E$ is located inside the cylinder $x^2 + y^2 = 1$ and between the circular paraboloids $z = 1 - x^2 - y^2$ and $z = x^2 + y^2$. Find the volume of $E$.

---

Volume $= \frac{\pi}{4}$

---

**Example 5: Multivariable Calculus.**

The graph of the polar rectangular region $D$ is given. Express the region $D$ in polar coordinates:

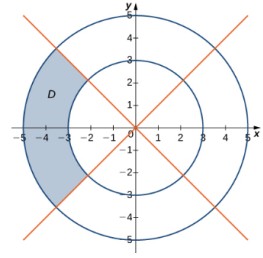

---

1. The interval of $r$ is $[3, 5]$
2. The interval of $\theta$ is $\left[\frac{3}{4} \cdot \pi, \frac{5}{4} \cdot \pi\right]$

---

**Example 6: Differential Calculus.**

Sketch the curve:

$$y = \frac{x^3}{6 \cdot (x + 3)^2}$$

Provide the following:
1. The domain (in interval notation)
2. Vertical asymptotes
3. Horizontal asymptotes
4. Slant asymptotes
5. Intervals where the function is increasing
6. Intervals where the function is decreasing
7. Intervals where the function is concave up
8. Intervals where the function is concave down
9. Points of inflection

---

1. The domain: $(-\inf, -3) \cup (-3, \inf)$
...

---

Figure 4: Example text-only and visual problems from the U-MATH benchmark, illustrating the topic, problem, and golden answer.

## A.2 U-MATH PROBLEM AND SOLUTION

### Example: Differential Calculus.

**U-MATH Problem:**
The function $s(t) = 2 \cdot t^3 - 3 \cdot t^2 - 12 \cdot t + 8$ represents the position of a particle traveling along a horizontal line.
1. Find the velocity and acceleration functions.
2. Determine the time intervals when the object is slowing down or speeding up.

**Golden answer:**
1. The velocity function $v(t) = 6 \cdot t^2 - 6 \cdot t - 12$ and acceleration function $a(t) = 12 \cdot t - 6$
2. The time intervals when the object speeds up $\left(0, \frac{1}{2}\right)$, $(2, \infty)$ and slows down $\left(\frac{1}{2}, 2\right)$

- - - - - - - - - - - - - - - - - - - - - - - - - - - - - - - - - - - - - - - - - - - - - - - - - - -

**Reference solution:**

The velocity is the derivative of the position function:

$$v(t) = s'(t) = \boxed{6t^2 - 6t - 12}$$

The acceleration is the derivative of the velocity function:

$$a(t) = v'(t) = \boxed{12t - 6}$$

To determine when the object is speeding up or slowing down, we compare the signs of $v(t)$ and $a(t)$.
***Step 1: Find the Zeros of*** $v(t)$ ***and*** $a(t)$
First, solve for $v(t) = 0$:

$$6t^2 - 6t - 12 = 0 \quad \Rightarrow \quad t^2 - t - 2 = 0 \quad \Rightarrow \quad (t-2)(t+1) = 0$$

Thus, $t = 2$ and $t = -1$.
Next, solve for $a(t) = 0$:

$$12t - 6 = 0 \quad \Rightarrow \quad t = \frac{1}{2}$$

***Step 2: Analyze the Signs of*** $v(t)$ ***and*** $a(t)$
We analyze the signs of $v(t)$ and $a(t)$ on the intervals determined by $t = -1$, $t = \frac{1}{2}$, and $t = 2$.

| Interval | $v(t)$ | $a(t)$ | Behavior |
|---|---|---|---|
| $(-\infty, -1)$ | $> 0$ | $< 0$ | Slowing down |
| $\left(-1, \frac{1}{2}\right)$ | $< 0$ | $< 0$ | Speeding up |
| $\left(\frac{1}{2}, 2\right)$ | $< 0$ | $> 0$ | Slowing down |
| $(2, \infty)$ | $> 0$ | $> 0$ | Speeding up |

***Step 3: Account for non-negative time***

The object is speeding up on $\boxed{\left(0, \frac{1}{2}\right) \text{ and } (2, \infty)}$ and slowing down on $\boxed{\left(\frac{1}{2}, 2\right)}$.

Figure 5: An example problem from the U-MATH benchmark, illustrating the problem, reference solution and golden answer.

## A.3  $\mu$-MATH META-EVALUATION

---

**Example: Integral Calculus.**

**U-MATH Problem:**
Solve the integral:

$$\int \frac{20 \cdot \cos(-10 \cdot x)^3}{21 \cdot \sin(-10 \cdot x)^7} \, dx$$

**Golden answer:**

$$C + \frac{1}{21} \cdot \left( \frac{1}{2} \cdot (\cot(10 \cdot x))^4 + \frac{1}{3} \cdot (\cot(10 \cdot x))^6 \right)$$

**LLM-generated answer:**

$$-\frac{3 \sin(10x)^2 - 2}{126 \sin(10x)^6} + C$$

- - - - - - - - - - - - - - - - - - - - - - - - - - - - - - - - - - - - - - - - - -

**Golden judge label:** ⟦correct⟧

**Comment:**
 The reference answer and the submitted one can be simplified, respectively, to

$$C + \frac{\cot^4(10x)}{42} + \frac{\cot^6(10x)}{63} \quad \text{and} \quad C + \frac{\cot^6(10x)}{63} + \frac{\cot^4(10x)}{42} + \frac{1}{126},$$

which differ by a constant term of $1/126$.

---

Figure 6: An example problem from the $\mu$-MATH meta-evaluation benchmark, illustrating the comparison between the golden (reference) answer and the answer generated by an LLM.

## B    SUB-TOPICS DISTRIBUTION

The U-MATH dataset cover variety of topics across 6 core subjects. Below is the count of unique topics per subject:

- Differential Calculus: 51 unique topics
- Sequences and Series: 28 unique topics
- Integral Calculus: 35 unique topics
- Precalculus Review: 19 unique topics
- Algebra: 74 unique topics
- Multivariable Calculus: 53 unique topics

| Subject | Topic Count | Topic Name |
|---|---|---|
| **Differential Calculus** | 29 | Curve Sketching |
| | 13 | Limits |
| | 12 | One-Sided Limits |
| | 12 | L'Hospital's Rule |
| | 11 | Increasing and Decreasing Functions |
| | 11 | Higher Derivatives |
| | 10 | Applications of Derivatives (Local Extrema) |
| **Sequences and Series** | 40 | Taylor Series |
| | 30 | Fourier Series |
| | 18 | Maclaurin Series |
| | 12 | Approximating Constants Using Power Series |
| | 6 | Radius of Convergence (Center of Convergence) |
| | 5 | Differentiate Power Series |
| | 4 | Error in Approximation |
| **Integral Calculus** | 83 | The Substitution Rule |
| | 24 | Antiderivatives |
| | 10 | Volumes of Solids of Revolution About the X-Axis |
| | 9 | Trigonometric Substitutions and Inverse Substitutions |
| | 9 | Integrate Respect Independent Variable |
| | 7 | Applications of Integrals |
| | 7 | Single Variable Surface Area Integrals |
| **Precalculus Review** | 55 | Trigonometric Functions |
| | 24 | Zeros |
| | 11 | Inverses of Functions |
| | 8 | Inequalities |
| | 7 | Equations with Exponents and Logarithms |
| | 7 | Properties of Functions |
| | 6 | Exponential Functions |
| **Algebra** | 18 | Equations and Inequalities |
| | 13 | Polynomial Equations |
| | 8 | Find Composition of Two Functions |
| | 7 | Polynomials |
| | 6 | Find Slope Line |
| | 6 | Applications of Exponential Function |
| | 6 | Quadratic Equations |
| **Multivariable Calculus** | 13 | Triple Integrals |
| | 11 | Lagrange Multipliers |
| | 9 | Double Integrals in Polar Coordinates |
| | 8 | Derivatives of Parametric Equations |
| | 8 | Integrals of Multivariable Functions |
| | 8 | Double Integral Over General Region |
| | 6 | Classification of Critical Points |

Table 6: Top 7 Topics for Each Subject.

## C PROMPTS

## C.1 PREDICTION PROMPT

> **Solution CoT Prompt.**
>
> {{problem}}\n
> Please reason step by step, and put your final answer within \ boxed{}
>
> - - - - - - - - - - - - - - - - - - - - - - - - - - - - - - - - - - - - - - - - - - -
>
> **Comment:**
> Images (if present) are passed with native for provider API schema. For OpenAI-compatible endpoints it is image_url field.[a]
> ___________________________
>
> [a]https://platform.openai.com/docs/guides/vision

Figure 7: Prediction for comparing student's answer and reference answer

## C.2 JUDGMENT PROMPT

---

**Judgment Automatic CoT Prompt.**

You'll be provided with a math problem, a correct answer for it and a solution for evaluation.
You have to answer whether the solution is correct or not.
---
PROBLEM STATEMENT:
{{problem}}
CORRECT ANSWER:
{{golden_answer}}
SOLUTION TO EVALUATE:
{{generated_solution}}
---
Now please compare the answer obtained in the solution with the provided correct answer to evaluate whether the solution is correct or not.
Think step-by-step, then conclude with your final verdict by putting either "Yes" or "No" on a separate line.

---

Figure 8: Judgment Automatic CoT Prompt for comparing student's answer and reference answer. This prompt has not been used in U-MATH evaluation.

---

**Judgment CoT Prompt.**

You'll be provided with a math problem, a correct answer for it and a solution for evaluation.
You have to answer whether the solution is correct or not.
---
PROBLEM STATEMENT:
{{problem}}
CORRECT ANSWER:
{{golden_answer}}
SOLUTION TO EVALUATE:
{{generated_solution}}
---
Now please compare the answer obtained in the solution with the provided correct answer to evaluate whether the solution is correct or not.
Think step-by-step, following these steps, don't skip any:
1. Extract the answer from the provided solution
2. Make any derivations or transformations that may be necessary to compare the provided correct answer with the extracted answer
3. Perform the comparison
4. Conclude with your final verdict — put either "Yes" or "No" on a separate line
---
For each question or part:
1. Write the reference answer and the student's final answer.
2. Make any derivations or transformations that may be necessary to compare the reference answer and the student's answer.
3. Only then perform the comparison.
After comparing all parts, provide a final judgment is the student's answer correct or incorrect.

---

Figure 9: Judgment CoT Prompt for comparing student's answer and reference answer. This is the prompt that has been used in U-MATH evaluation.

---

**Judgment Extract Prompt.**

You'll be given a result of an evaluation of some mathematical solution by a professional evaluator. You need to extract the final verdict of this evaluation in simple terms: is the solution graded as correct or not.
Output only a single label — "Yes", "No" or "Inconclusive" — according to the provided evaluation ("Yes" if the solution is graded as correct, "No" if the solution is graded incorrect, "Inconclusive" if the evaluation is incomplete or the final verdict is not settled upon).
Only output "Inconclusive" for incomplete or unsettled evaluations. If the evaluation does contain a single final verdict like "Yes", "Correct", "True", "No", "Incorrect", "False" and so on, even if it is supplied with some additional disclaimers and remarks, output a "Yes" or "No" label accordingly.

Here goes your input:
```
{{generated_judgemnt}}
```

Now please output exactly either "Yes", "No" or "Inconclusive".

---

Figure 10: Prompt for extracting the final verdict from the judge's outputs.

## D  SOLUTION PREDICTIONS LENGTH DISTRIBUTION

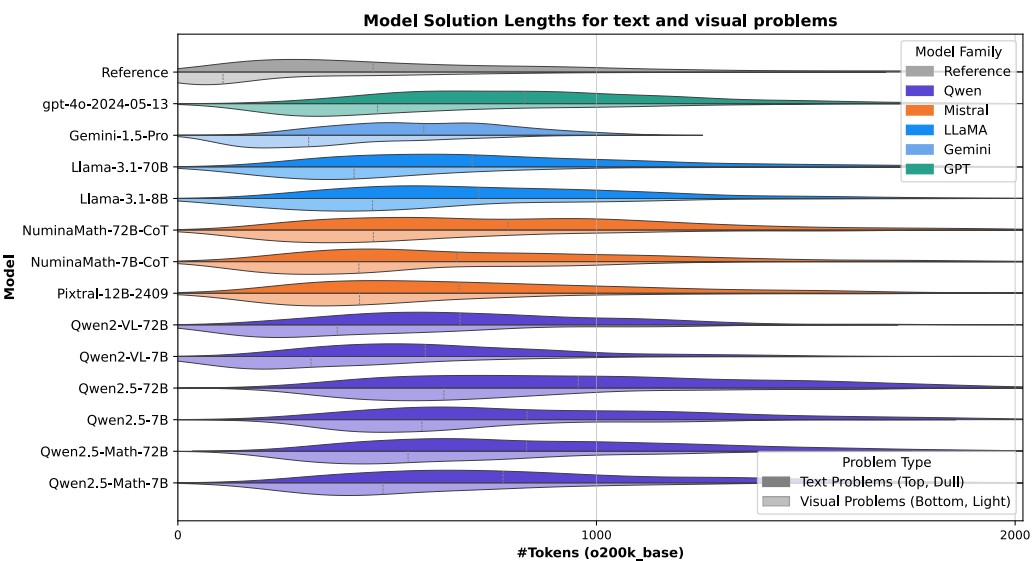

Figure 11: Distribution of token number for generated solutions: Text-only problems (top, dull) and Visual problems (bottom, light). `o200k_base` tokenizer is used for consistency.

# E MODEL ACCURACY VS SIZE

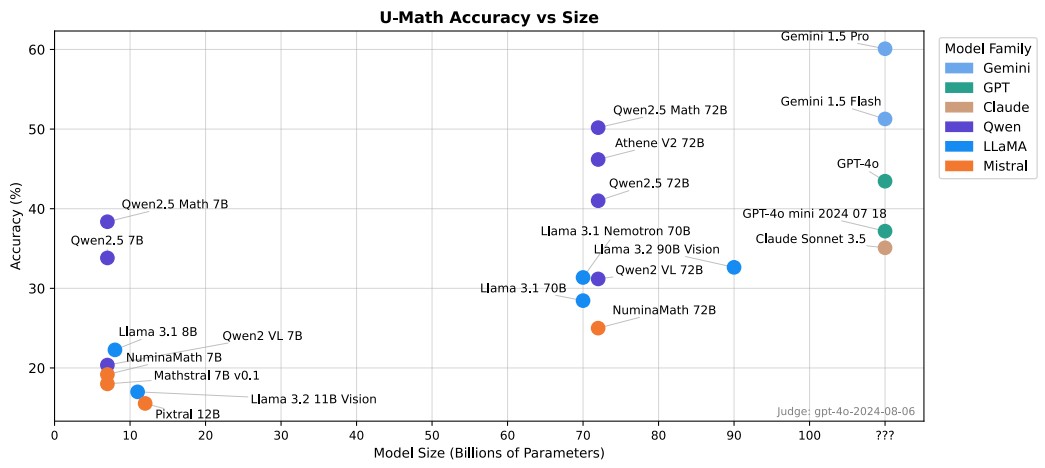

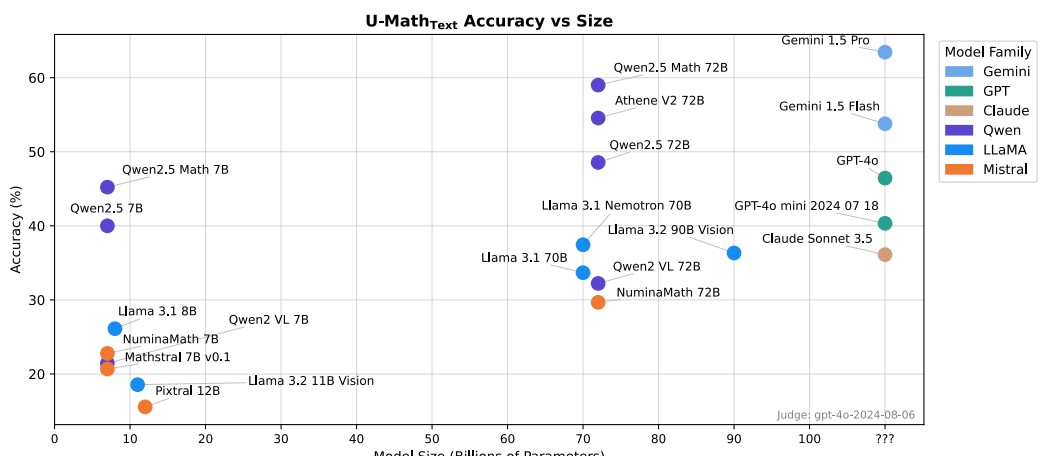

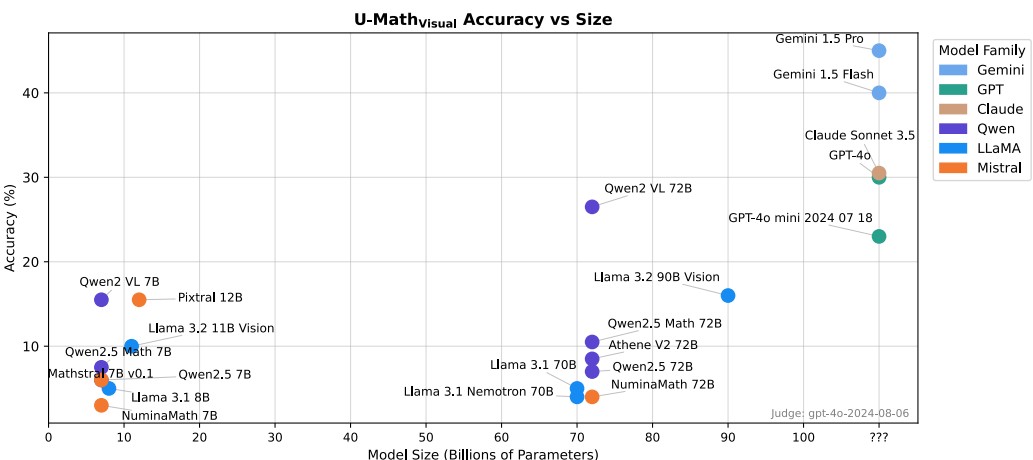

Figure 12: Accuracy of the selected top-performing models on U-MATH, U-MATH_Text, and U-MATH_Visual. Color denotes different model families. Higher is better for all charts.

# F  $\mu$-MATH PROMPTING SCHEMES

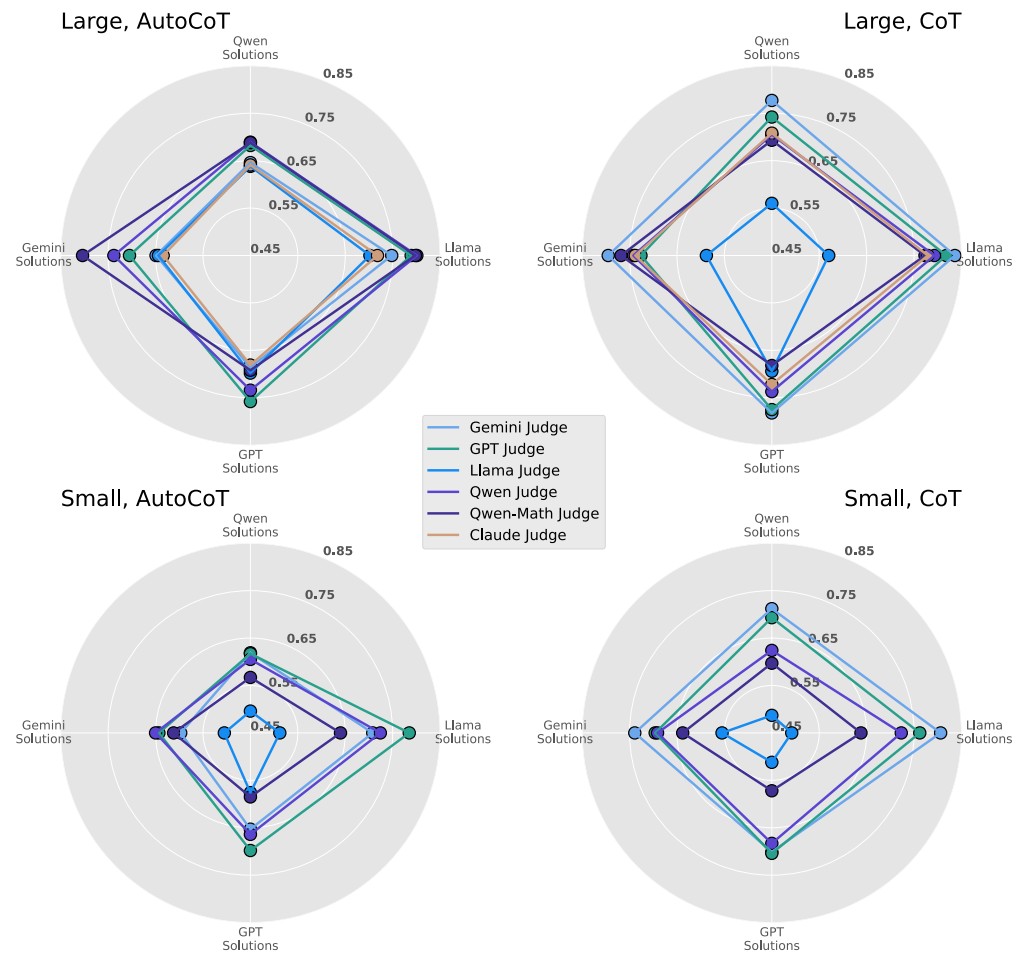

Figure 13: $\mu$-MATH macro F1-score for each of the four solution author models, split by judge model size and prompting scheme.

# G   $\mu$-MATH RATES OF INCONCLUSIVE JUDGMENTS

| | Llama-3.1 8B | LLaMA-3.1 70B | Qwen2.5-Math 7B | Qwen2.5-Math 72B | Qwen2.5 7B | Qwen2.5 72B | GPT-4o-mini | GPT-4o | Gemini 1.5 Flash | Gemini 1.5 Pro | Claude 3.5 Sonnet |
|---|---|---|---|---|---|---|---|---|---|---|---|
| **Automatic CoT** | 13.4 | 5.0 | 2.8 | 1.2 | 1.0 | 1.6 | 0.0 | 0.0 | 0.0 | 0.0 | 0.0 |
| **CoT** | 22.9 | 13.8 | 2.4 | 0.7 | 1.2 | 2.1 | 0.1 | 0.0 | 0.1 | 0.0 | 0.0 |

Table 7: Percentages of inconclusive judgments produced by each model under different prompting schemes.

## H  COMPARISON OF PROBLEM SOLVING AND JUDGMENT PERFORMANCE

In this section, we provide a detailed comparison of model performance on U-MATH and . The overall distribution of scores visualized in Figure 14 not only shows that improved problem-solving performance does not immediately lead to better judgment performance, as discussed in Section 4.3, but also suggests a possible trade-off existing between these capabilities.

This possibility is further illustrated when considering specific models. For instance, the Qwen2.5-Math model demonstrates strong problem-solving compared to most of the models, but does so at the expense of weaker instruction following — eye-gaze inspections reveal the model struggling with instruction comprehension and adherence to formatting rules — leading to a lower judgment performance relative to others. In contrast, Claude does not rank low as a judge despite its weak performance on U-MATH. Meanwhile Gemini, known, to excel in both mathematical problem-solving and instruction following, comes out as the top-ranked judge.

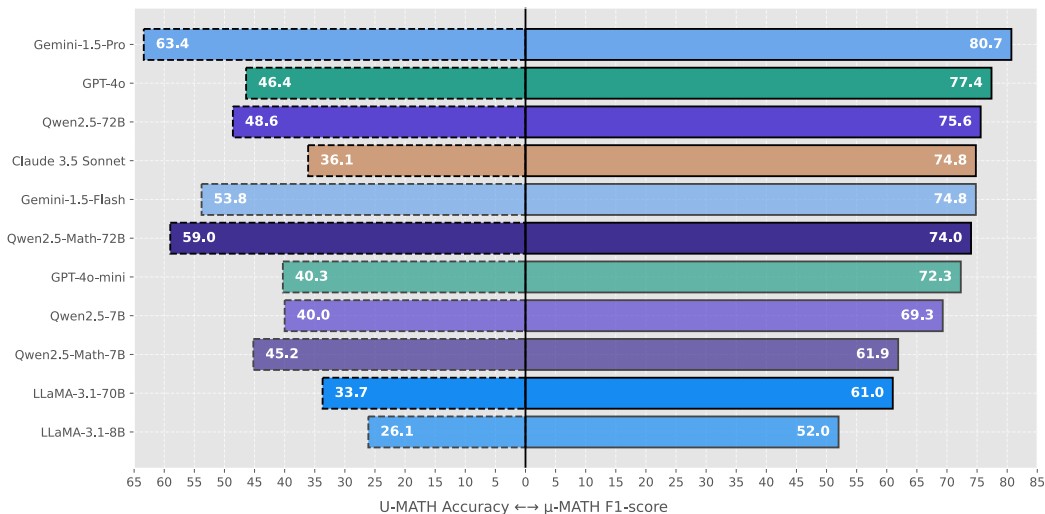

Figure 14: Comparison of problem solving () vs judgment performances ($\mu$-MATH) of the various models. Higher is better in both cases.

# I  μ-MATH μ-MATH BEHAVIOR OF JUDGES

In Figure 15 we visualize the difference in 'performance profiles' among the judges observed discussed in Section 4.3 — proprietary models being more conservative and Qwen family models exhibiting the opposite tendencies.

The difference in behavior patterns may also be observed in predicted label agreement rates between judges, see Figure 16 for the comparison. Interestingly, **no pair of models has agreement above around 80%**, even for same-family models like Qwen2.5 and Qwen2.5-Math, despite the pairwise μ-MATH performance diffs being small compared to 20% disagreement. This shows that judge comparison is substantive beyond the one-dimensional choice of the better model and suggests judge ensembling to be a potentially fruitful approach to evaluation.

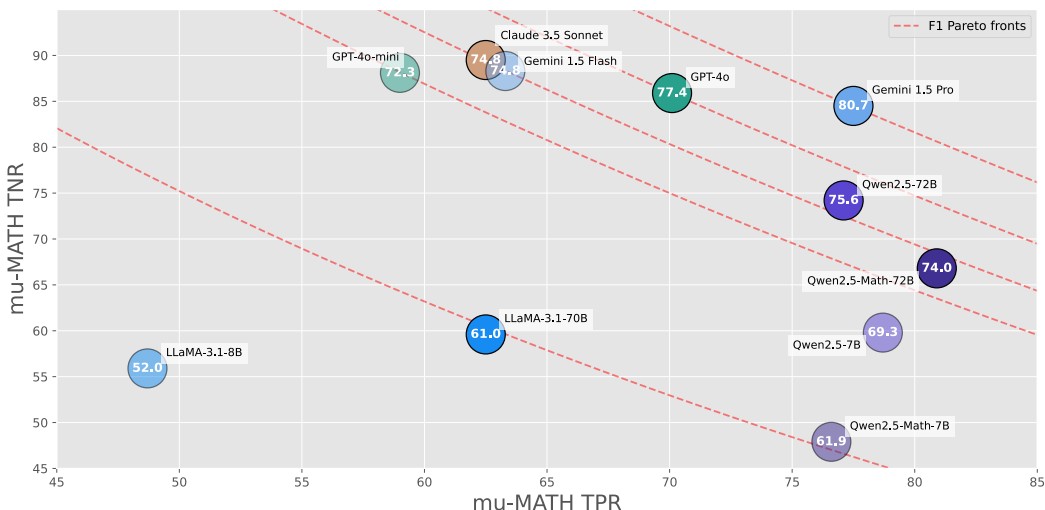

Figure 15: True Positive Rate vs True Negative Rate of judges in μ-MATH. The value inside of the marker denotes the macro F1-score.

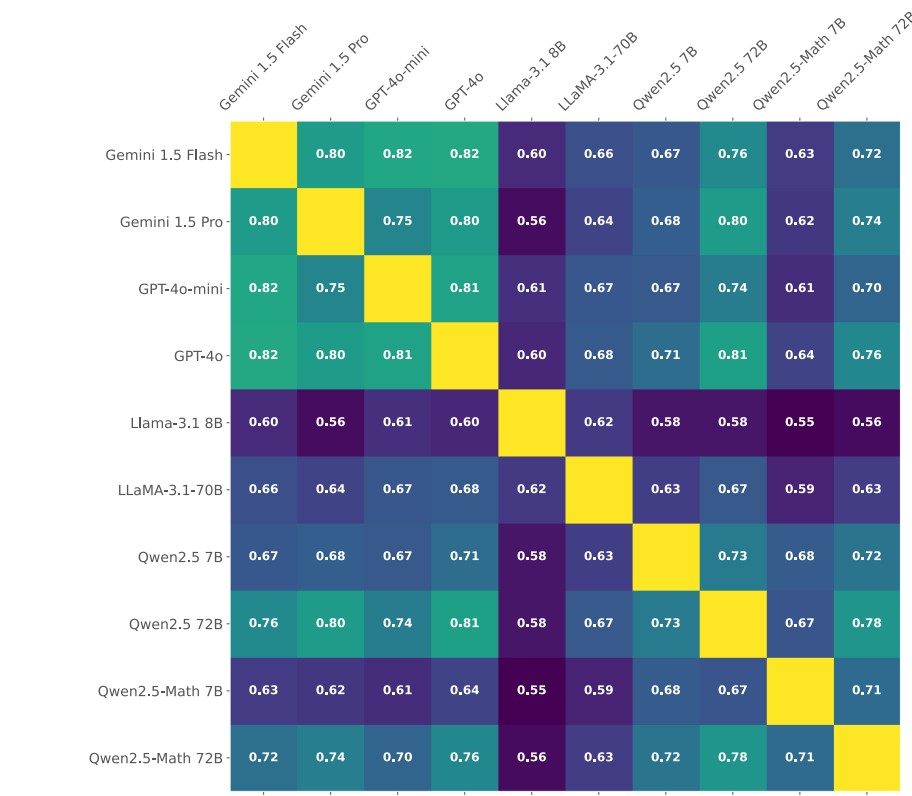

Figure 16: Agreement between different judges on $\mu$-MATH as measured by predicted label coincidence ratio.

## J   FULL $\mu$-MATH RESULTS

| Model | $\mu$-MATH | | | | | $\mu$-MATH$_{Qwen}$ | | | | | $\mu$-MATH$_{Llama}$ | | | | | $\mu$-MATH$_{GPT}$ | | | | | $\mu$-MATH$_{Gemini}$ | | | | |
|---|---|---|---|---|---|---|---|---|---|---|---|---|---|---|---|---|---|---|---|---|---|---|---|---|---|
| | F1 | TPR | TNR | PPV | NPV | F$_{macro}$ | TPR | TNR | PPV | NPV | F1 | TPR | TNR | PPV | NPV | F1 | TPR | TNR | PPV | NPV | F1 | TPR | TNR | PPV | NPV |
| Llama-3.1 8B | 52.0 | 52.3 | 48.7 | 55.9 | 56.0 | 48.7 | 49.3 | 45.2 | 53.4 | 56.5 | 49.2 | 49.7 | 45.4 | 54.0 | 35.5 | 51.2 | 51.5 | 46.4 | 56.5 | 53.3 | 55.5 | 57.9 | 55.0 | 62.2 | 77.0 |
| Qwen2.5 7B | 69.3 | 69.7 | 78.7 | 59.8 | 69.3 | 62.4 | 62.8 | 75.5 | 49.1 | 66.5 | 72.3 | 73.3 | 78.4 | 70.1 | 59.4 | 68.3 | 69.3 | 81.4 | 55.7 | 66.3 | 69.1 | 69.2 | 79.4 | 59.8 | 82.0 |
| Qwen2.5-Math 7B | 61.9 | 62.8 | 76.6 | 47.9 | 62.9 | 59.7 | 59.8 | 71.0 | 48.3 | 64.7 | 63.8 | 68.4 | 85.6 | 51.7 | 49.7 | 57.2 | 58.6 | 75.0 | 41.2 | 57.7 | 63.8 | 63.8 | 77.8 | 50.0 | 78.2 |
| LLaMA-3.1 70B | 61.0 | 61.0 | 62.5 | 59.6 | 64.1 | 56.0 | 56.0 | 58.7 | 53.4 | 62.8 | 57.0 | 58.6 | 63.9 | 54.0 | 43.7 | 69.4 | 69.4 | 67.1 | 71.8 | 71.8 | 58.8 | 60.3 | 61.4 | 61.0 | 78.4 |
| Qwen2.5 72B | 75.6 | 75.6 | 77.1 | 74.2 | 77.5 | 70.5 | 70.5 | 76.1 | 64.7 | 74.2 | 79.3 | 79.3 | 75.3 | 83.9 | 72.3 | 73.7 | 73.8 | 76.4 | 71.0 | 73.8 | 74.2 | 74.5 | 79.4 | 72.0 | 86.7 |
| Qwen2.5-Math 72B | 74.0 | 74.2 | 80.9 | 66.8 | 73.8 | 69.3 | 69.8 | 81.3 | 56.9 | 71.6 | 77.3 | 77.9 | 81.4 | 76.4 | 65.8 | 68.2 | 68.7 | 77.9 | 58.8 | 66.9 | 76.8 | 77.0 | 82.5 | 73.2 | 87.6 |
| GPT-4o-mini | 72.3 | 74.3 | 59.0 | 88.1 | 85.1 | 69.3 | 72.1 | 56.1 | 87.1 | 85.3 | 76.2 | 76.8 | 59.8 | 90.2 | 77.3 | 70.4 | 72.2 | 56.4 | 86.3 | 81.4 | 69.6 | 73.3 | 63.0 | 87.8 | 92.2 |
| GPT-4o | 77.4 | 78.1 | 70.1 | 85.9 | 85.1 | 74.2 | 75.2 | 67.1 | 83.6 | 84.6 | 81.8 | 82.0 | 71.1 | 90.8 | 81.2 | 77.5 | 77.8 | 72.1 | 83.2 | 82.1 | 72.6 | 74.6 | 70.4 | 82.9 | 90.5 |
| Gemini 1.5 Flash | 74.8 | 76.3 | 63.3 | 88.3 | 86.2 | 71.2 | 73.1 | 60.6 | 85.3 | 84.7 | 80.6 | 81.1 | 67.0 | 92.0 | 82.3 | 70.1 | 71.6 | 57.9 | 84.0 | 79.4 | 73.9 | 77.2 | 67.7 | 91.5 | 94.8 |
| Gemini 1.5 Pro | 80.7 | 80.9 | 77.5 | 84.5 | 85.2 | 77.7 | 78.0 | 75.5 | 81.0 | 84.2 | 83.6 | 83.7 | 75.3 | 90.8 | 82.0 | 78.2 | 78.3 | 76.4 | 80.2 | 80.5 | 79.5 | 80.2 | 81.0 | 82.9 | 91.6 |

Table 8: Comparison of model's ability to judge on $\mu$-MATH benchmark, with CoT prompting; Macro F1-score (F1), True Positive Rate (TPR), True Negative Rate (TNR), Positive Predictive Value (PPV), and Negative Predictive Value (NPV), with F1 as the primary one are presented. Columns under $\mu$-MATH represent the integral score over the entire benchmark, while $\mu$-MATH $_{<model>}$ is subset with solutions generated by specific model.**Bold** indicates the best result for each column. Reference full table in Appendix J.

