# OpenReview forum: "U-MATH: A University-Level Benchmark for Evaluating Mathematical Skills in LLMs"
_ICLR.cc/2025/Conference — Submitted to ICLR 2025_

### Official Review · Reviewer_9qXr · 2024-11-01

**Soundness:** 2
**Presentation:** 2
**Contribution:** 2
**Rating:** 5
**Confidence:** 4

**Summary:**

The paper introduces U-MATH, a novel benchmark designed to evaluate the mathematical reasoning capabilities of Large Language Models (LLMs) at the university level. It comprises 1,125 unpublished, open-ended problems sourced from actual teaching materials, balanced across six core mathematical subjects, with 20% of the problems requiring image understanding. Additionally, the paper presents µ-MATH, a meta-evaluation dataset aimed at assessing the ability of LLMs to evaluate free-form mathematical solutions. The experiments conducted reveal significant challenges in advanced mathematical reasoning and visual problem-solving, with the best-performing models achieving only 53% accuracy on text-based tasks and 30% on visual problems. The paper also highlights the difficulty LLMs face in assessing solutions, with the highest µ-MATH F1-score being 76%, indicating room for improvement in LLMs’ evaluation capabilities. The datasets and evaluation code are open-sourced to facilitate further research.

**Strengths:**

The paper demonstrates a high-quality collection of problems that are well-balanced across six core mathematical subjects. This ensures a comprehensive evaluation of LLMs across different areas of mathematics.
The problems sourced from actual teaching materials add a layer of authenticity and practical relevance to the benchmark, ensuring that the skills assessed are applicable to real-world academic standards.
The creation of µ-MATH for meta-evaluation is an innovative approach to assessing the ability of LLMs to evaluate mathematical solutions. This adds another layer of complexity and originality to the benchmarking process, focusing not just on problem-solving but also on the assessment capabilities of the models.

**Weaknesses:**

While the inclusion of visual elements in 20% of the problems is a step forward, the remaining 80% are text-based. The paper could benefit from expanding the visual problem set to better assess and train LLMs in multimodal mathematical reasoning, which is increasingly important for real-world applications.

The paper focuses on university-level mathematics, but it is unclear how well the findings generalize to other levels or types of mathematical reasoning. Future work could explore the transferability of the models trained on U-MATH to other mathematical domains.

**Questions:**

Why are there no examples of problems that require visual input?

The accuracy when using LLM as a judge is not provided, especially for higher mathematics problems where answers may be in different forms but are actually equivalent, indicating that it is easier to make mistakes compared to comparing a single form of answer.

---

> ### Author Response · Authors · 2024-11-15
> **Response to Reviewer 9qXr**
>
> 9qXr comments upon the proportion of visual problems in the U-MATH dataset (20% of problems with image). While we recognize the importance of multimodal reasoning, we aim to strike a balance between text and visual input. Choosing 20% of the problems to be visual we mirror the distribution of problems given to students in top-tier US universities to align with real-world academic expectations.
>
> 9qXr finds it interesting to explore the generalization of models trained on university-level problems. As we emphasize in the general comment, the primary goal of this study is to introduce a robust benchmark for post-hoc evaluation of LLM capabilities. However, this question aligns with our future research ideas regarding samples we did not select for the benchmark. We will make the existing Future Work paragraph in Section 5 more clear to state it. Also, we include a comparison of the performance of existing models across different benchmarks in Appendix XXX.
>
> 9qXr also notes a lack of examples of problems with visual input. We improve the presentation of benchmarks by adding more diverse examples for U-MATH in Appendix A.
>
> 9qXr points out that accuracy for LLM judging is not provided. In Section 4.2.2 we analyze the performance of different models-as-a-judges and report F1 score along with Precision and Recall. We choose the F1 measure for the μ-MATH benchmark to minimize the effect of the class imbalance present, as we discuss in Section 3.3.

---

### Official Review · Reviewer_rZHQ · 2024-11-03

**Soundness:** 2
**Presentation:** 2
**Contribution:** 2
**Rating:** 5
**Confidence:** 3

**Summary:**

The authors introduced a new benchmark dataset, U-MATH, designed to evaluate large language models (LLMs) on university-level math problems. The proposed U-MATH benchmark includes 1,125 college-level math problems collected from real educational materials, covering six core mathematical subjects, with 20% of the problems involving image understanding. Additionally, the paper introduces a meta-evaluation dataset named µ-MATH, aimed at assessing the ability of LLMs to judge the correctness of mathematical solutions.

**Strengths:**

1.U-MATH Benchmark: This is a publicly available dataset of university-level math problems, covering six topics: Pre-Calculus, Algebra, Differential Calculus, Integral Calculus, Multivariable Calculus, and Sequences & Series. A unique aspect of this dataset is its inclusion of open-ended questions that require LLMs to perform multi-step reasoning.
2.µ-MATH Meta-Evaluation Benchmark: This benchmark is specifically designed to test LLMs’ ability to assess the correctness of mathematical solutions. It contains 340 questions selected from U-MATH, accompanied by LLM-generated answers manually labeled as correct or incorrect, aimed at evaluating the capacity of LLMs to act as “judges.”
3.Model Comparison: The paper compares the performance of various LLMs, including general-purpose models, specialized math models, and multimodal models, demonstrating the significant challenges LLMs still face in both text and visual tasks. For instance, the highest accuracy for text-based questions is 53%, while performance on visual questions is even lower, with an accuracy of only 30%.
4.Challenges for LLMs as Math Judges: LLMs perform poorly when evaluating mathematical solutions, with the best-performing LLM judge achieving an F1 score of only 76% on µ-MATH, indicating that there is still room for improvement in this task.

**Weaknesses:**

1.The U-MATH dataset introduced in the paper supplements the current math datasets by addressing college-level gaps, while the µ-MATH meta-evaluation dataset enables assessment of large models’ ability to evaluate college-level math solutions. However, aside from knowing that this training set focuses on university mathematics and includes six subjects, we lack information about the dataset’s question diversity, difficulty, reasoning steps required to solve the problems, and other aspects. Additionally, the dataset’s size may be insufficient.
2.The paper mentions that the dataset has been released but does not provide an access link, so I have no direct way to review the dataset.
3.The experiments in the paper provide valuable insights into the capabilities of current text-based and multimodal LLMs in solving university-level math problems.
4.The paper states that U-MATH aims to promote further research and improve LLMs' ability to handle complex math problems. How is "complex" defined here? Does it refer to higher-grade, more challenging (for humans) knowledge, or does it mean problems requiring more and deeper reasoning steps?

**Questions:**

I don't have further questions.

---

> ### Author Response · Authors · 2024-11-15
> **Response to Reviewer rZHQ**
>
> rZHQ mentions a lack of information about the dataset diversity. We include the distribution of sub-topics in Appendix B to show up diversity and depth of the dataset.
>
> rZHQ requests the access link to the dataset. We add an anonymous download link to the current dataset version.
>
> rZHQ expresses concern regarding the use of the term “complex”. We define “complex” in this study as problems both of higher education level, so requiring deeper knowledge as well as a higher number of steps, compared to existing datasets. We will clarify this definition in the paper to avoid ambiguity.

---

### Official Review · Reviewer_Zva3 · 2024-11-04

**Soundness:** 2
**Presentation:** 3
**Contribution:** 3
**Rating:** 6
**Confidence:** 3

**Summary:**

The paper introduces U-MATH, a comprehensive university-level mathematical benchmark designed to evaluate the performance of Large Language Models (LLMs) in solving advanced mathematical problems. The dataset consists of 1,125 problems sourced from university coursework, covering six core topics such as Algebra, Calculus (Differential and Integral), Multivariable Calculus, Sequences, and Series, with approximately 20% of the tasks involving visual components. To complement the U-MATH dataset, the authors also present µ-MATH, a meta-evaluation set for assessing the accuracy and reliability of LLM-based evaluators.

**Strengths:**

S1. The inclusion of university-level problems offers a significant advancement over existing datasets that mainly focus on elementary or high school-level tasks.

S2: By integrating visual tasks alongside traditional textual ones, the dataset challenges LLMs to interpret and reason across multimodal formats.

S3: µ-MATH introduces a novel approach to evaluate LLMs' ability to assess solutions, addressing biases and limitations in current evaluation practices.

**Weaknesses:**

W1: The reliance on LLMs as judges (e.g., GPT-4o) to evaluate free-form answers could introduce biases and inconsistencies, particularly since LLMs may struggle with complex derivations or nuanced interpretations of mathematical expressions.

W2: The µ-MATH set includes LLM-generated solutions, which may limit the diversity and challenge of evaluation due to inherent model tendencies or training biases. This could result in less rigorous meta-evaluation as models may overfit to known patterns or heuristics.

**Questions:**

What measures have been taken to mitigate potential biases introduced by using LLMs as judges for solution correctness?

---

> ### Author Response · Authors · 2024-11-15
> **Response to Reviewer Zva3**
>
> Zva3 makes a point that using LLM-as-a-judge for free-form answers introduces bias.
> 1. An alternative option would be to use multiple-choice format for answers. While this format reduces judgment error, it simplifies the problem for a model and can lead to guessing, thereby reducing the complexity and authenticity of the benchmark (W.Li et.al, 2024; P.Pezeshkpour et.al, 2023). Moreover, developing “plausible” alternative options, especially in high numbers as used in MMLU-Pro (Y.Wang et.al, 2024), is a complex task, requiring professors to use exactly this problem in examinations before and collect students’ mistakes.
> 2. To address and measure judgment biases on complex derivations we design µ-MATH (Section 3.3), providing a meta-evaluation of LLM’s judging capabilities. µ-MATH allows us to directly measure the reliability of LLMs in assessing solutions and to account for any inconsistencies introduced by the judging model.
> 3. Also, as discussed in Section 4.2.2, we select the most “conservative” model (GPT-4o) for all the judging. Recognizing that, while any model can introduce bias, the overall ranking of models should remain consistent.
>
> Zva3 points out that µ-MATH relies on LLM-generated solutions which may add bias. It is important to emphasize that this benchmark’s purpose is to assess models as evaluators for other LLMs, rather than human students, making it crucial to include generated solutions in the dataset. However, we are actively working on extending µ-MATH to include solutions from multiple model families and add human-made adversarial samples, which should enhance evaluation diversity further. Although, the µ-MATH dataset does include stylized human-generated solutions with the correct label. We will clearly state it in the paper.
>
> Zva3 asks about measures taken to mitigate LLM-as-a-Judge bias introduced. First of all, we select the judge which yields the highest True Positive Rate (TPR) and Positive Predictive Value (PPV), shown in Table 5. Additionally, we compose robust judging prompts with examples embedded to minimize derivations and ensure CoT reasoning (Appendix C.2).
>
> References:
> Wang, Yubo, et al. "Mmlu-pro: A more robust and challenging multi-task language understanding benchmark." arXiv preprint arXiv:2406.01574 (2024).
> Li, Wangyue, et al. "Can multiple-choice questions really be useful in detecting the abilities of LLMs?." arXiv preprint arXiv:2403.17752 (2024).
> Pezeshkpour, Pouya, and Estevam Hruschka. "Large language models sensitivity to the order of options in multiple-choice questions." arXiv preprint arXiv:2308.11483 (2023).

---

> > ### Comment · Reviewer_Zva3 · 2024-12-02
> >
> > I find your rebuttal comprehensive and satisfactory. While challenges remain, your responses clearly articulate the current limitations and reasonable future plans for addressing them. Based on this, I will maintain my original score for the paper.

---

### Official Review · Reviewer_H6kh · 2024-11-04

**Soundness:** 2
**Presentation:** 3
**Contribution:** 2
**Rating:** 5
**Confidence:** 4

**Summary:**

This paper introduces the U-Math datasets, based on university-level mathematics, addressing the issues of insufficient thematic diversity and a lack of visual information question types in current datasets for evaluating the mathematical abilities of large language models. The U-Math datasets was tested on several large language models, revealing that the highest accuracy for text-based tasks was only 53%, while the highest accuracy for visual tasks was only 30%.

**Strengths:**

1. The paper is well-organized, providing a clear outline of the datasets, experimental setup, and evaluation metrics. The authors explain each component in a structured manner, making it accessible to readers.

2. The datasets include a range of mathematical subjects and problem types, which reflects an effort to cover diverse aspects of mathematical reasoning, though the depth and breadth could still be improved.

3. The introduction of U-MATH and µ-MATH provides additional benchmarks for evaluating LLMs in mathematical tasks, which may offer a reference point for similar studies.

**Weaknesses:**

1. Although the U-MATH datasets consists of 1,125 samples and covers six subjects, the sample size is still too small. Evaluating the mathematical abilities of large models using a limited amount of data is not sufficiently convincing.

2. Although the 340 samples in the µ-MATH datasets have been carefully selected to provide a challenging test, a larger sample size could enhance the representativeness of the evaluation, especially across different topics and problem types.

**Questions:**

1.It is recommended to expand both the U-MATH datasets size and the number of subjects.

2.It is recommended to expand the µ-MATH datasets size.

3.In Table 4, you only use accuracy to present the results. Since the study involves math problems, which are more complex than simple classification tasks, could you consider adding additional evaluation metrics like perplexity or WinoGrande ACC (to assess whether ambiguous problems are correctly identified)? This would give readers a clearer picture of how well the models truly understand and respond to university-level math questions. For more details, you might refer to examples in this paper: https://proceedings.mlr.press/v235/dao24a.html.

---

> ### Author Response · Authors · 2024-11-15
> **Response to Reviewer H6kh**
>
> H6kh expresses concern regarding the depth and breadth of the benchmark. During data collection and validation we ensure that the problems reflect actual academic curricula of top-tier US universities. We include the distribution of sub-topics in Appendix B.
>
> We appreciate H6kh’s concern regarding the dataset size. As mentioned in the general comment, the U-MATH is the largest university-level math benchmark available (Table 1). Moreover, with 1,125 samples the margin of error is ±3.1% (at a 95% confidence level, with p=50% of model being correct) for model accuracy estimation, given the statistically significant difference for most of the models in Table 4.
>
> We strongly agree with H6kh’s suggestion to increase the µ-MATH dataset size. As mentioned in the general comment, we are actively working on expanding not only the size to 700–900 samples but on improving the diversity of the benchmark. By diversifying the solution source models, we aim to increase both the depth and representativeness of µ-MATH.
>
> H6kh finds it interesting to explore alternative evaluation metrics. We choose to use accuracy, as it is the primary evaluation metric used to evaluate solutions free from mathematical tasks (D.Hendrycks et.al, 2021;  J.Ahn et.al, 2024, M.Fang et.al, 2024). However, we agree that other metrics can be useful for more fine-grained analysis, such as perplexity or step-by-step solution evaluations, which can lead to more fine-grained evaluation of LLMs. As we address in Section 5, we plan to investigate this in the following papers.
>
> References:
> Hendrycks, Dan et al. “Measuring Mathematical Problem Solving With the MATH Dataset.” ArXiv abs/2103.03874 (2021): n. Pag.
> Ahn, Janice, et al. "Large language models for mathematical reasoning: Progresses and challenges." arXiv preprint arXiv:2402.00157 (2024).
> Fang, Meng, et al. "Mathodyssey: Benchmarking mathematical problem-solving skills in large language models using odyssey math data." arXiv preprint arXiv:2406.18321 (2024).

---

> > ### Comment · Reviewer_H6kh · 2024-11-26
> >
> > Thank you for the detailed response. After carefully reviewing the authors’ response and the revised manuscript, several key issues remain unresolved:
> >
> > 1. The dataset is described as reflecting the academic curricula of top-tier US universities, with the distribution of sub-topics provided in Appendix B. While the effort is acknowledged, the dataset still lacks sufficient diversity in visual problems and advanced mathematical topics. These limitations hinder a comprehensive evaluation of mathematical reasoning abilities in academic scenarios, potentially affecting the generalizability of the results.
> >
> > 2. The explanation regarding the margin of error and the assertion that μ-MATH is the largest available university-level mathematics benchmark are appreciated. However, the dataset size of 1,125 samples appears insufficient to fully capture the breadth of university-level mathematics, which spans significantly broader and more complex topics compared to K-12 curricula. This limitation raises concerns about the reliability of the evaluation. While the margin of error is noted as ±3.1% at a 95% confidence level (assuming a correctness rate of 50%), the statistical methods or parameters used to derive this figure are not detailed. Providing a thorough explanation of the statistical methodology would enhance the transparency and credibility of the findings.
> >
> > 3. The intention to expand the μ-MATH dataset and improve its diversity to increase its depth and representativeness is acknowledged. However, these improvements are still in the planning stage without concrete results, which limits the credibility of the commitment. Including a detailed plan for dataset expansion, with specific objectives, timelines, and anticipated outcomes, would strengthen the manuscript.
> >
> > 4. Accuracy is stated as the primary evaluation metric, with future plans to explore more granular metrics such as perplexity or step-by-step solution evaluations. While the potential of these directions is recognized, relying solely on accuracy in the current study fails to capture the nuanced performance of large language models in complex mathematical reasoning tasks. Incorporating additional evaluation metrics in this submission would enhance its rigor and applicability.
> >
> > In light of the above considerations, I will keep the score.

---

### Author Response · Authors · 2024-11-15
**General response**

We thank the reviewers for their helpful comments. In this revision, we:
* Improve the presentation by adhering to English style and grammar; we additionally use the premium Grammarly subscription for post-editing;
* Add more examples of the problems to the Appendix A, including visual problems (9qXr);
* Provide distribution of sub-topics for 6 code subjects in Appendix B to showcase the depth and diversity of the benchmark (H6kh, Zva3);
* Add an anonymous download link to U-MATH and µ-MATH datasets in Abstract (rZHQ).

Following valuable feedback, in the next revision we will:
* Increase the number of µ-MATH samples while also extending the diversity of the dataset by including solutions generated by various model families. We aim to enhance the depth and robustness of the benchmark, ensuring it accurately reflects the broader challenges encountered during solution evaluation (H6kh)
* Investigate statistics on the number of solution steps required for solution in addition to the existing number of questions/answers per problem (Zva3);
* Add additional models to the comparison: gpt-4o-mini, Gemini-1.5-Flash, LLaMA-3.2, and Athene-V2 to strengthen both depth and width of our research

The following points are mentioned by several reviewers, so we address them here:
* It is important to note, that the primary goal of this work is to introduce a robust and comprehensive benchmark for post-hoc evaluation of LLMs’ math capabilities in university-level settings. So we do not provide a training set for U-MATH and µ-MATH benchmarks; they are presented solidly for evaluations of LLMs. (Zva3, 9qXr)
However, reviewers’ interest in exploring the generalizability of models trained on university-level mathematical problems to other educational levels aligns with ours. We also recognize the value of researching new methods to enhance the training process, particularly for advanced mathematics. We plan to investigate these aspects in future work.
* Regarding dataset size, to our knowledge, U-MATH is the largest university-level math benchmark available (Table 1).
However, we aim to provide an affordable way of testing models; validation of larger sample sizes would increase costs, as demonstrated by the [MathEval dataset](https://github.com/math-eval/MathEval ), which includes 30k samples but authors spent around $58,000 for API calls.
Moreover, the current sample size of 1,125 yields a confidence interval with the highest margin of error being ±3.1% (at a 95% confidence level, with p=50% of a model being correct) for model accuracy estimation, which we deem statistically robust for a benchmark of this scale, particularly given a gap exceeding 3% between the top-scoring and second-ranking models. The closest dataset OCWCourses yields a ±5.9% Margin of Error with 272 samples. Moreover, 150 samples in each text-only subject split and 225 samples in the visual split provide statistically significant differences between most models in ranking.

---

### Author Response · Authors · 2024-11-28
**General Responce - v2 revision**

We further update the paper following the reviewer's comments. In this revision, we:
* We **substantially update the µ-MATH dataset**, expanding its size to 1084 solutions generated by a diverse set of models: Qwen2.5 72B, LLaMA-3.1 70B, GPT-4o and Gemini 1.5 Pro. This expansion not only increases statistical robustness, but allows fine-grained analysis of model-judgeing itself to analyse bias (Zva3).
* We also provide more detailed analyzes of judges’ performance, prompt sensitivity, behavior patterns and biases, better showcasing the utility of meta-evaluations.
* We include additional models to comparison: Gemini Flash, GPT-4o-mini, Claude 3.5 Sonnet, LLaMA-3.2 11B/90B, LLaMA-3.1 Nemotron 70B, and Athene-V2 72B, improving depth of our analysis and highlighting ‘further-tuning’ improvement for Nemotron and Athene.
* We update scores of the models using a more standardized prediction prompt used in DeepSeek (A.Liu et.al, 2024) and Qwen-2.5-Math (A.Yang et.al, 2024) models, and a simpler, better-performing judgment prompt, with double the inference token limit.
* We also update the U-MATH dataset, to increase the interlap with µ-MATH problems. This together with changes in prompting setups for prediction and judgment lead to changes in final scores on U-MATH. However, the models’ ranking remains stable, with Gemini being the top performer.
* Also, we improve presentation of U-MATH and µ-MATH results by including additional charts in the paper body and Appendix sections, allowing for a detailed comparison of the models’ performance across various splits.

---

### Meta-Review · Area_Chair_1ZTE · 2024-12-21

**Metareview:**

This paper presents U-MATH, a university-level mathematical benchmark containing 1,125 problems for evaluating LLMs, along with μ-MATH, a meta-evaluation dataset for assessing solution correctness. While the paper makes a valuable contribution by addressing the gap in university-level math assessment, significant concerns remain about the limited dataset size, potential biases in LLM-based evaluation, and lack of comprehensive analysis of problem complexity. Although the authors provided detailed responses and planned improvements, core issues around statistical robustness and evaluation methodology were not fully resolved. The dataset's current scope and evaluation approach require substantial enhancement before it can serve as a reliable benchmark for advanced mathematical reasoning capabilities in LLMs.

**Additional Comments On Reviewer Discussion:**

The reviewers (H6kh, Zva3, 9qXr) consistently rated the paper as borderline rejection, citing concerns about dataset size, evaluation methodology, and potential biases. While authors addressed some issues in their revision - adding topic distribution details, visual problem examples, and planning μ-MATH expansion - fundamental concerns about statistical robustness and evaluation biases remain unresolved. H6kh's critiques about dataset limitations and evaluation metrics were particularly noteworthy. Although authors argued for statistical significance with ±3.1% margin of error, the lack of detailed statistical methodology and limited problem diversity undermines the benchmark's reliability for comprehensive LLM evaluation in university-level mathematics.

---

### Decision · Program_Chairs · 2025-01-22

Reject